# QTG-Miner aids rapid dissection of the genetic base of tassel branch number in maize

Xi Wang [1,2,6], Juan Li[1,2,6], Linqian Han[1,2], Chengyong Liang[1,2], Jiaxin Li[1,2], Xiaoyang Shang[1,2], Xinxin Miao[1,2], Zi Luo[1,2], Wanchao Zhu[1,2], Zhao Li[1,2], Tianhuan Li[1,2], Yongwen Qi[3], Huihui Li [4], Xiaoduo Lu[5] & Lin Li [1,2] ✉

Genetic dissection of agronomic traits is important for crop improvement and global food security. Phenotypic variation of tassel branch number (TBN), a major breeding target, is controlled by many quantitative trait loci (QTLs). The lack of large-scale QTL cloning methodology constrains the systematic dissection of TBN, which hinders modern maize breeding. Here, we devise QTG-Miner, a multi-omics data-based technique for large-scale and rapid cloning of quantitative trait genes (QTGs) in maize. Using QTG-Miner, we clone and verify seven genes underlying seven TBN QTLs. Compared to conventional methods, QTG-Miner performs well for both major- and minor-effect TBN QTLs. Selection analysis indicates that a substantial number of genes and network modules have been subjected to selection during maize improvement. Selection signatures are significantly enriched in multiple biological pathways between female heterotic groups and male heterotic groups. In summary, QTG-Miner provides a large-scale approach for rapid cloning of QTGs in crops and dissects the genetic base of TBN for further maize breeding.

Genetic improvement of agronomic traits is an efficient way to produce high-yield and high-quality crop varieties and ensure global food security. Most agronomic traits are mainly controlled by multiple minor-effect quantitative trait loci (QTLs), which are often involved in complex genetic regulatory networks[1–3]. Compared to most functional genes identified based on their associated mutants with pleiotropic or extreme phenotypes, causal genes underlying QTLs often exhibit more moderate phenotypic variation, and the favorable alleles in natural populations hold great value in the genetic improvement of crop plant architecture, yield and quality[1,3]. Therefore, the rapid cloning and functional dissection of quantitative trait genes (QTGs) are of great importance in the genetic improvement of crops, which can help feed the world population.

Map-based cloning and genome-wide association studies (GWAS) are powerful tools for cloning QTGs[1,4–9] and have enabled dissection of the genetic mechanisms behind important agronomic traits[3,8–11]. However, both methods have some limitations in terms of operability and efficiency. Map-based cloning largely depends on the construction of mapping populations with a large number of recombinants whose genotype and phenotype are assessed for fine mapping, which is time-consuming and labor-intensive[1,12]. GWAS is powerful in detecting common alleles in diverse genetic populations, but is more limited for rare alleles, as it must reach a minimal allele frequency within the population to be detectable[8,13,14]. Complex population structures also affect the identification of functional genetic loci[8,13,14]. More robust and large-scale genetic methods are therefore desired for modern maize breeding.

[1]National Key Laboratory of Crop Genetic Improvement, Huazhong Agricultural University, Wuhan 430070, China. [2]Hubei Hongshan Laboratory, Wuhan 430070, China. [3]College of Agriculture and Biology, Zhongkai University of Agriculture and Engineering, Guangzhou 510325 Guangdong, China. [4]Institute of Crop Science, Chinese Academy of Agricultural Sciences, 100081 Beijing, China. [5]Institute of Molecular Breeding for Maize, Qilu Normal University, Jinan 250200, China. [6]These authors contributed equally: Xi Wang, Juan Li. ✉e-mail: hzaulilin@mail.hzau.edu.cn

With the development of various omics technologies, the acquisition of multi-dimension omics data is becoming cheaper and easier. Multi-omics data provide an unprecedented chance for functional gene cloning and dissection of the genetic mechanisms behind important agronomic traits in crops[15]. Walley et al. constructed a large-scale integrated gene expression atlas composed of messenger RNAs, nonmodified proteins, and phosphoproteins quantified across maize development, which was effective in predicting known and novel regulatory relationships[16]. Using a systems biology approach, Clark et al. integrated multi-omics datasets, unraveled the molecular signaling events of the brassinosteroid (BR) response in Arabidopsis (*Arabidopsis thaliana*), and verified the involvement of transcription factor BRONTOSAURUS (BRON) in regulating cell division modulated by BR-responsive kinases and transcription factors[17]. Utilization of multi-omics networks is therefore powerful for functional gene prediction. Nevertheless, how to integrate multi-omics data and traditional QTL mapping for the rapid fine mapping and cloning of QTGs remains elusive and unclear.

One of the most widely cultivated crops, maize (*Zea mays* ssp. *Mays*) contributes over 38% of the world's cereal production[18]. High-density planting due to the compact plant architecture of maize has largely driven increasing global production[19]. Compact plant architecture is associated with reduced relative ear height, smaller tassels, and fewer tassel branch numbers (TBNs). During maize domestication and improvement, the tassel morphology of maize has changed significantly. For instance, tassel size and TBN have diminished, which directly affects light interception and grain yield[11,19]. Dissecting the genetic architecture of TBN is important for the further genetic improvement of plant architecture and maize yield. Cloning of conventional TBN genes based on isolation of their corresponding mutants has enabled the identification of a series of TBN genes and uncovered several molecular pathways behind TBN[20–29]. However, only a few of the causal genes underlying QTLs have been cloned, even though several hundred QTLs have been identified[30,31], which limits the pace of modern maize improvement.

Here, we devise QTG-Miner, a multi-omics data-based method for large-scale and efficient fine mapping and cloning of QTGs in maize. We apply QTG-Miner to 12 TBN QTLs simultaneously and successfully clone seven TBN QTLs. Selection analysis shows that *lrs1* (*liguleless-related sequence1*) has been subjected to selection during modern maize improvement. Additionally, we assemble a comprehensive molecular regulatory network underlying TBN in maize and uncover that significant co-directional selection signatures are enriched at multiple biological pathways between female heterotic groups (FHGs) and male heterotic groups (MHGs). Our study provides an approach for rapidly cloning QTLs and systematically dissecting the molecular mechanisms underlying TBN in maize, which could be helpful for the genetic improvement of this and other important agronomic traits in crops.

## Results

### Rationale behind QTG-Miner for large-scale and rapid fine mapping and cloning of QTLs

For rapid fine mapping and cloning of QTLs on a large scale, we devised QTG-Miner, which consists of three steps (Fig. 1).

(1) Primary QTL mapping. Segregating populations can be constructed using two parents (P1 and P2) showing distinct phenotypes. We devised QTG-Miner to accept various populations, which include but are not limited to F$_2$, doubled haploids (DHs), recombination inbred line (RILs), and near isogenic line (NILs). Primary QTL mapping can be conducted by any method of the user's choice based on linkage analysis and association mapping, for example, QTL IciMapping[32], WinQTLCart[33] or R/qtl[34].

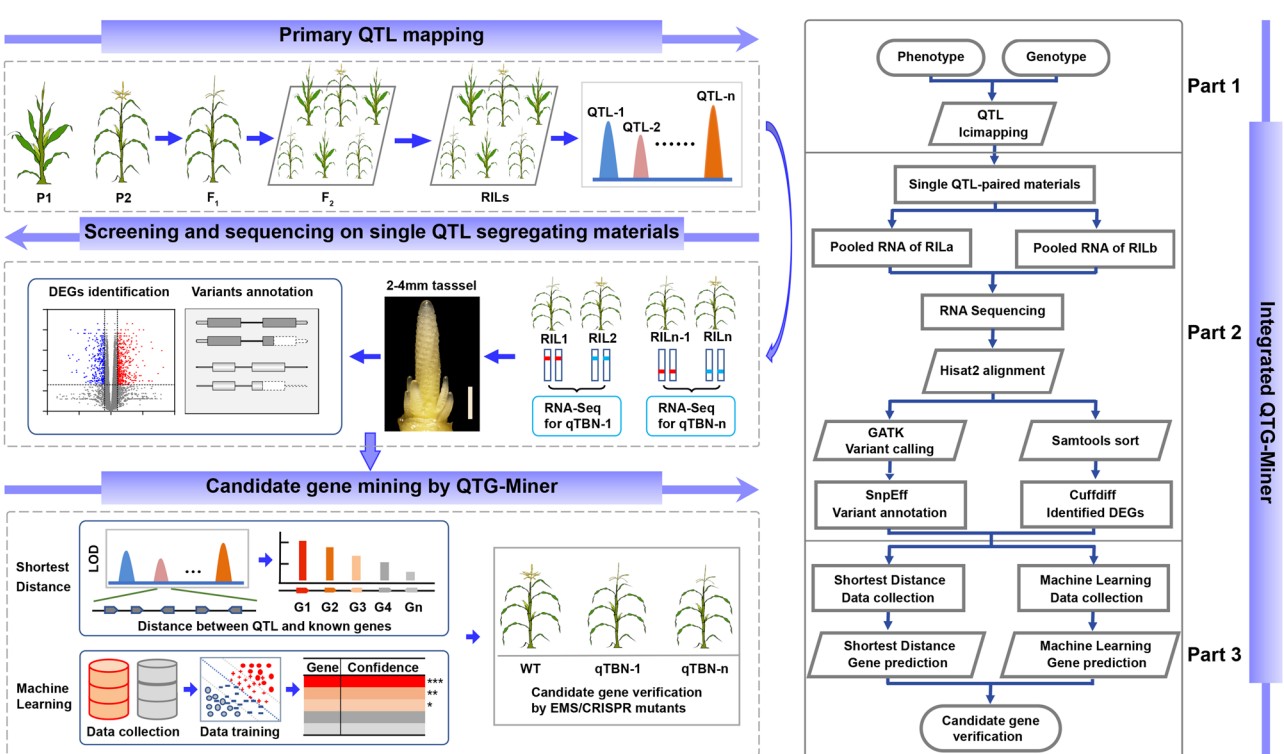

**Fig. 1 | Rationale behind QTG-Miner.** QTG-Miner integrates three procedures: 1) primary QTL mapping, 2) screening and sequencing on single QTL segregating material, and 3) candidate gene mining by QTG-Miner. In 1), mapping population types include but are not limited to F$_2$, DHs, RILs, and NILs. In 2), single QTL-paired materials are screened, planted, and sampled for RNA-seq. Identified DEGs and sequence variants are considered as proportional weights during candidate gene prediction. In 3), SD and ML algorithms are integrated into QTG-Miner. Using SD and ML algorithms, candidate genes underlying the target QTL can be uncovered and subsequently verified by EMS mutagenesis or CRISPR/Cas9-mediated editing.

(2) Screening and sequencing on single QTL segregating material. Based on the QTLs identified in part 1, individuals from the $F_2$ segregating, RIL, or NIL populations that harbor different genotypes across the target QTL interval while being as close as possible genetically outside of this region are then characterized for the identification of differentially expressed genes (DEGs) and sequence variants by transcriptome deep sequencing (RNA-seq).

(3) Candidate gene mining by QTG-Miner. QTG-Miner integrates multi-omics network maps, artificial intelligence, and interpretation of the biological consequences of mutations or polymorphisms detected between the contrasting individuals above. We implemented two algorithms in QTG-Miner, shortest distance (SD) and machine learning (ML), based on the multi-omics network maps. The SD algorithm involves acquisition of a dataset of positive (known trait-associated) genes, followed by calculation of the SD between genes located within the QTL interval and these positive genes, and leading to the identification of candidate genes. By contrast, ML relies on acquisition of a dataset of positive (known trait-associated) and negative (verified trait-unassociated) genes, model training, and candidate gene prediction. Several commonly used ML algorithms such as bagging, extreme gradient boosting (XGBoost), logistic regression (LR), NeuralNet and support vector machine (SVM) were introduced into QTG-Miner[35]. The best model can be determined by various metrics, such as receiver operating characteristic (ROC), area under the curve (AUC) and area under the precision-recall curve (AUPRC). During candidate gene prediction, identified DEGs and genic sequence variants can be prioritized by giving them a proportionally greater weight than non-DEGs or genes with no sequence polymorphisms between individuals under consideration. Based on the prediction results and functional annotations of all genes in the target QTL interval, a high-confidence candidate gene can be selected. Thereafter, mutagenesis of the candidate gene can be explored in an existing ethyl methanesulfonate (EMS) population or via clustered regularly interspaced short palindromic repeat (CRISPR)/CRISPR-associated nuclease 9 (Cas9)-mediated editing to verify the candidate gene underlying the target QTL (Fig. 1).

QTG-Miner therefore integrates QTL mapping, screening and sequencing of materials segregating only at the target QTL, and candidate gene mining, which should accelerate fine mapping and cloning of functional genes conferring important agricultural traits. We set out to challenge QTG-Miner to identify genes involved in TBN below.

## QTG-Miner rapidly narrows down candidate genes underlying 12 TBN QTLs in maize

TBN is an important agronomic trait in maize production and breeding. To dissect the genetic basis of TBN and to test the reliability of QTG-Miner, we applied QTG-Miner simultaneously to 12 maize TBN QTLs, which had been previously mapped in 10 RIL populations derived from 14 diverse elite inbreds[36]. These 12 TBN QTLs map to six different chromosomes (2, 3, 4, 7, 8, and 10) and can explain 4–14% of the standing phenotypic variation in TBN (Supplementary Data 1). Based on the genotype of each RIL, we identified 12 pairs of RILs for the 12 TBN QTLs with opposite genotypes over a single QTL of interest, but with a very similar genomic background outside the QTL interval (Supplementary Data 1). The detailed genotype information of 12 pairs of RILs were exhibited and graphically presented (Supplementary Data 2–7, Supplementary Fig. 1). We sowed all lines in a field and collected developing tassels of 2–4 mm in length from each RIL for RNA-seq analysis (Supplementary Data 8). We quantified the expression of genes and annotated the genic sequence variants in all lines over each specific target QTL interval, resulting in 12 lists of DEGs and genes with polymorphisms (Supplementary Data 9 and 10). For sequence variants, we focused on 13 types of variants with a high probability of altering gene expression and/or protein structure or length: conservative in-frame deletion, conservative in-frame insertion, disruptive in-frame deletion, disruptive in-frame insertion, frameshift, splice acceptor,

splice donor, splice region, lost start codon, gained stop codon, lost stop codon, 5_prime_UTR_variant and 3_prime_UTR_variant. We incorporated these identified DEGs and major-effect variants as proportional weights during candidate gene mining by QTG-Miner.

With the rapid development of various sequencing methods, an integrative multi-omics network map was assembled (Fig. 2a) and evidenced to accelerate the dissection of biological pathways and prediction of gene function[37], which integrated multi-omics data from ChIA-PET (chromatin interaction analysis by paired-end tag sequencing), co-expression, co-translation and PPI (protein-protein interaction). This multi-omics network map had not yet been applied for the fine mapping and cloning of QTLs.

In this study, we integrated the above multi-omics network map into QTG-Miner for the application of QTG mapping. We collected 57 known functional genes affecting TBN as positive genes and 63 negative genes with no evidence of being connected to TBN (Fig. 2b, Supplementary Data 11). Based on the integrative network map, we calculated the SD value ($SD_g$) between each gene across the maize genome and the positive genes. The mean $SD_g$ between 52 positive genes and 52 positive genes was significantly higher than that between 26,044 background genes and 62 negative genes (Fig. 2c). Using five ML algorithms, we conducted model training and predictions and determined that NeuralNet exhibits the best prediction accuracy, with a mean AUC of 0.93 compared to the other four algorithms (Fig. 2d, Supplementary Table 1). We therefore used NeuralNet to predict the candidate genes underlying each of the 12 TBN QTLs. We separately extracted the prediction results of SD and ML for the 12 TBN QTLs, treating the identified DEGs and major-effect variants as co-factors. We narrowed down the QTL regions and prioritized the functional candidate genes underlying these 12 TBN QTLs (Fig. 2e, f).

## Validation of functional genes underlying seven TBN QTLs in maize

To validate the candidate genes identified by QTG-Miner, we used EMS materials and CRISPR-edited materials for phenotyping validation (Supplementary Data 12 and 13). Of the 12 TBN QTLs, we successfully cloned and functionally analyzed seven TBN candidate genes underlying seven TBN QTLs (Fig. 3, Supplementary Fig. 2). Taking *qTBN3-1* and *qTBN7-1* as examples, the candidate gene *Zm00001d042795* was ranked as the top candidate by both SD and ML algorithms (Fig. 3b, c). Sequencing analysis of the two parental alleles (B73 and BY804) at *qTBN3-1* locus detected one polymorphism of type stop_gained (CAG > TAG), resulting in a premature translation termination codon (Supplementary Fig. 3). *Zm00001d042795* gene encodes a protein with a kinesin motor domain, ZmKinesin. In Arabidopsis, a knockout mutant in the homologous gene *FRAGILE FIBER 1* (*FRA1*) exhibited a dwarf plant stature and a small inflorescence architecture[38]. Compared to their wild-type sibling counterparts, EMS mutants exhibited significantly increased tassel branch numbers (Fig. 3d, e). Meanwhile, two independent CRISPR-edited materials were phenotyped, and also showed significant increase of tassel branch number, which further verified the causal gene of *qTBN3-1* (Fig. 3f–h). For *qTBN7-1*, the candidate gene *Zm00001d020804* was ranked as the top and third candidate by both SD and ML algorithms respectively (Fig. 3j, k). Sequencing analysis of the two parental alleles (DE3 and BY815) at this locus detected one polymorphism of 8-bp InDel, as well as differential expressions (Supplementary Fig. 3, Supplementary Data 9). *Zm00001d020804* gene encodes the homeobox leucine zipper domain transcription factor ZmHD-ZIP120. CRISPR-edited materials were phenotyped, and showed significant decrease of tassel branch number, which verified the causal gene of *qTBN7-1* (Fig. 3l–n).

We observed significant variation in TBN phenotypes in all mutants compared to their wild-type sibling counterparts. Mutant plants for both *ZmHD-ZIP120* and *ZmPRP4K* (*Pre-mRNA processing 4 KINASE*, the candidate gene for *qTBN8-3*) had significantly decreased TBN, while the mutants of the remaining five genes exhibited significantly increased

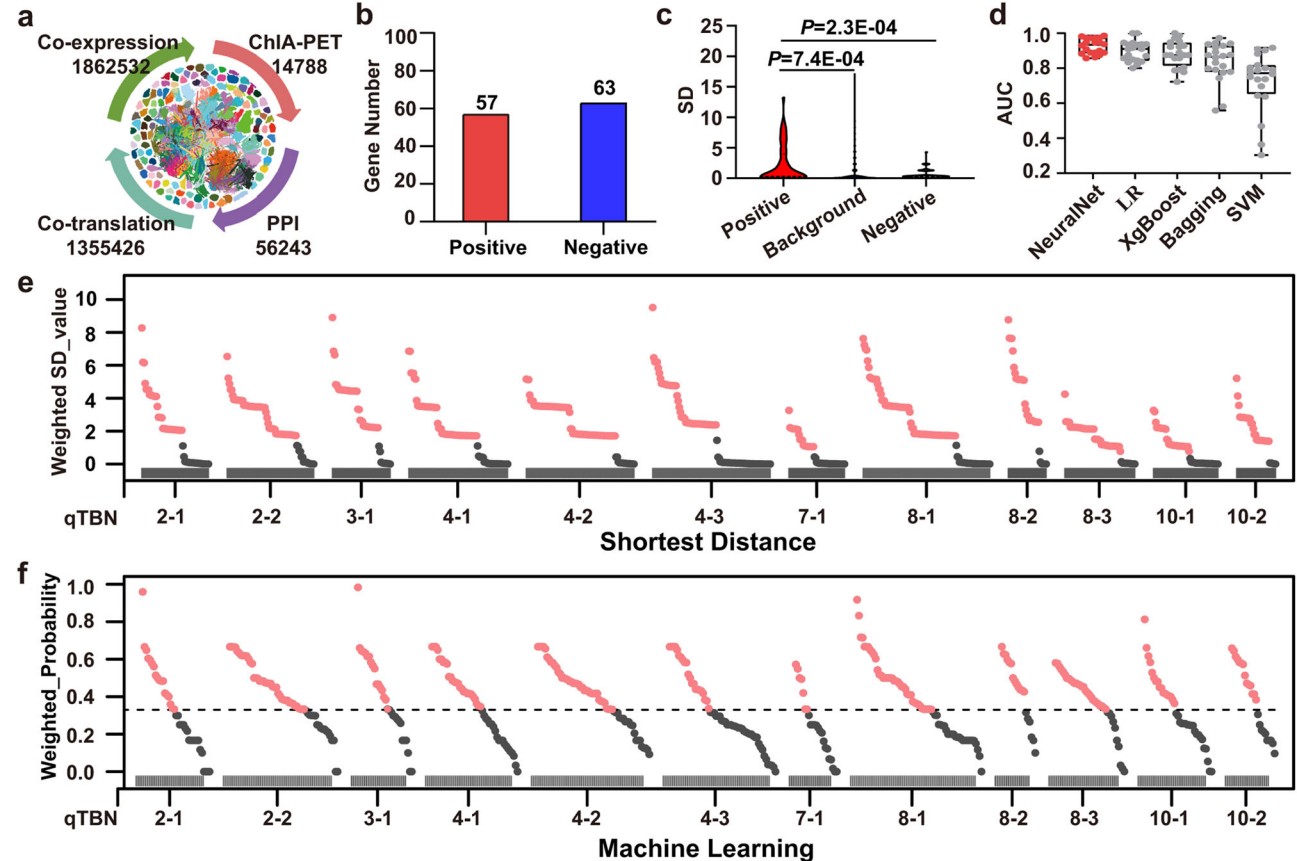

**Fig. 2 | QTG-Miner quickly narrows down candidate genes underlying 12 TBN QTLs in maize. a** Multi-omics network map used by QTG-Miner. ChIA-PET, chromatin interaction analysis by paired-end tag sequencing. PPI, protein-protein interaction. **b** Positive and negative gene datasets used for the SD and ML algorithms. **c** Comparison of SD values from positive genes ($n = 52$), background genes ($n = 26,044$) and negative genes ($n = 62$) to positive genes. The y-axis indicates the SD value. Violin plots: median ± upper and lower quartiles; $P$ values were calculated from two-sided Student's $t$ tests. **d** AUC values obtained from five different ML algorithms. For five algorithms, $n = 20$ independent replicates, respectively. In each box plot, the center line indicates the median, the edges of the box represent the first and third quartiles, and the whiskers extend to span a 1.5 interquartile range from the edges. **e** Weighted SD value ($SD_w$) of each gene in the candidate interval for 12 TBN QTLs. **f** Weighted probability ($P_w$) of each gene in 12 TBN QTLs, gray dotted line indicates the cutoff of machine learning. In **e**, **f**, Salmon solid dots indicate potential candidate genes, and gray solid dots indicate excluded genes.

TBN across multiple environments. These results confirmed that the genes prioritized by QTG-Miner are the genes underlying the TBN QTLs (Fig. 3, Supplementary Fig. 2). We conclude that we validated a high proportion of candidate genes identified by QTG-Miner.

Further, we summarized the success rate of cloning of TBN QTLs with different effects. In field tests from at least two seasons/locations, 7 of the 12 TBN candidate genes highlighted by QTG-Miner were validated (Supplementary Fig. 4a). To test the performance of QTG-Miner for minor-effect QTLs, we classified the seven verified TBN QTLs based on the strength of their effects. These seven TBN QTLs have logarithm of the odd (LOD) values ranging from 3.3 to 8.4 and effects of 4.6–14.4%. Of the 12 TBN QTLs, 5 QTLs whose LOD values below 5, and 2 QTLs whose LOD values larger than 5, were successfully verified, respectively (Supplementary Fig. 4b). Taken together, these results indicate that QTG-Miner exhibited good performance for the cloning of TBN QTLs with both major and minor effects.

**A comprehensive molecular network underlying TBN in maize**
To systematically dissect the molecular mechanisms underlying TBN, we utilized the multi-omics network map to decipher the molecular network of TBN in maize, which consisted of 65 well-known TBN genes and seven cloned genes from this study (Fig. 4a, Supplementary Data 14). To construct the molecular network underlying TBN, we performed RNA-seq analyses on the EMS mutants of six of the seven

cloned TBN genes and their wild-type sibling counterparts (Supplementary Data 8). Totally, we identified 1349 DEGs, which are potentially involved in the TBN network (Supplementary Data 15). Additionally, we conducted transient and simplified Cleavage Under Targets and Tagmentation (tsCUT&Tag)[39] on the protein encoded by the TBN gene, the homeobox leucine zipper domain transcription factor ZmHD-ZIP120 (the causal gene for *qTBN7-1*), integrated the multi-omics gene regulatory network constructed in our previous study for the dissection of the regulatome of TBN genes[40], and identified 957 potential target genes totally (Supplementary Fig. 5, Supplementary Data 16). The integration of all information generated a large-scale molecular network describing TBN in maize. This TBN network included 21,280 interaction edges derived from 4278 genes (Fig. 4a). Based on numerous published studies, the genes from the total TBN network could be roughly divided into 12 biological pathways (abscisic acid [ABA] and reactive oxygen species [ROS], auxin, boundary, brassinosteroid, cytokinin, cytoskeleton and cellulose, flowering, gibberellin, histone modification, meristem maintenance and determinacy, protein modification and transport, sugar and nutrition). Gene Ontology (GO) enrichment analysis showed that these 4278 genes are significantly enriched in 1247 terms, which included these 12 biological pathways (Fig. 4b, Supplementary Data 17).

We aimed to infer the genetic basis of TBN in maize through exploiting the assembled TBN network mentioned above. For

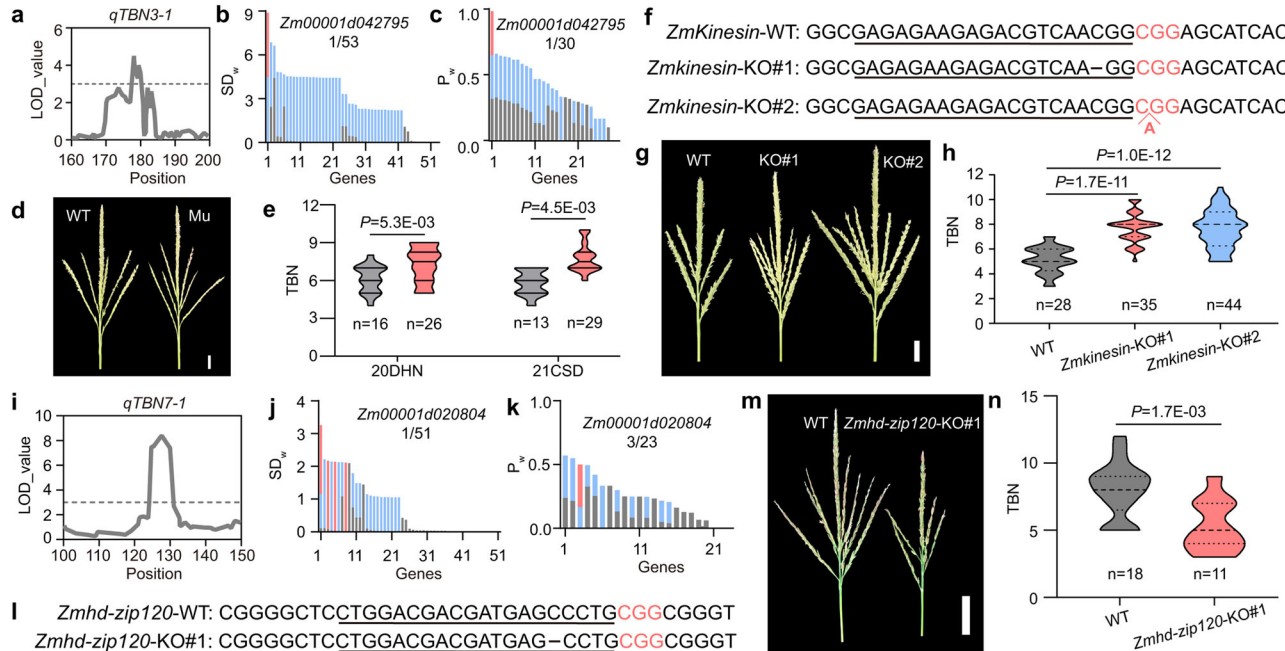

**Fig. 3 | Verification of candidate genes identified by QTG-Miner for *qTBN3-1* and *qTBN7-1* in maize. a** Primary genetic mapping results of *qTBN3-1*. **b** Detailed weighted SD values ($SD_w$) of genes in the interval of *qTBN3-1*. **c** Detailed weighted_Probability ($P_w$) of genes in the interval of *qTBN3-1*. **d** Representative photographs showing the TBN phenotype in wild type (left) and mutant *Zmkinesin-EMS* (right) derived from EMS materials. **e** TBN in wild type (gray violin plots) and mutants (pink violin plots). 20DHN, Hainan in winter 2020. 21CSD, Shandong in spring 2021. **f** Sequences of *ZmKinesin* target regions in wild type, *Zmkinesin-KO#1* and *Zmkinesin-KO#2* CRISPR/Cas9 knockout mutants. **g** Representative photographs showing the TBN phenotype in wild type (left), *Zmkinesin-KO#1* (middle) and *Zmkinesin-KO#2* (right) mutants. **h** TBN in wild type (left), *Zmkinesin-KO#1* (middle) and *Zmkinesin-KO#2* (right) mutants. **i** Primary genetic mapping results of *qTBN7-1*. **j** Detailed weighted SD values ($SD_w$) of genes in the interval of *qTBN7-1*.

**k** Detailed weighted_Probability ($P_w$) of genes in the interval of *qTBN7-1*. **l** Sequences of *ZmHD-ZIP120* target regions in wild type, *Zmhd-zip120-KO#1* CRISPR/Cas9 knockout mutants. **m** Representative photographs showing the TBN phenotype in wild type (left) and *Zmhd-zip120-KO#1* (right) mutant. **n** TBN in wild type (left) and *Zmhd-zip120-KO#1* (right) mutant. In **b** and **j**, Pink indicates $SD_d$, light blue indicates $SD_v$, and gray indicates $SD_g$. In **c** and **k**, Pink indicates $P_d$, light blue indicates $P_v$, and gray indicates $P_g$. In **f**, **l**, the target sites and protospacer-adjacent motifs (PAM) are shown as underscored letters and pink letters, respectively. The gap lengths of sequences are shown above the wild type sequences. Scale bars referred above; 2 cm. In **e**, **h**, **n**, *P* values were determined by two-sided Student's *t* tests. **P* < 0.05, ***P* < 0.01, ****P* < 0.001. Violin plots: median ± upper and lower quartiles. Source data are provided as a Source Data file.

example, we constructed a subnetwork for *qTBN8-2* (*lrs1*), which comprised 91 interaction pairs derived from the multi-omics network map, a transcription factor (TF)-centered gene regulatory network (GRN) and DEGs identified using EMS-mutagenized materials (Fig. 4c). We noticed several important genes in the resulting network, such as *tasselsheath4* (*tsh4*), *NIGHT LIGHT-INDUCIBLE AND CLOCK-REGULATED GENE 1* (*LNK1*), *REVEILLE 1* (*RVE1*), *MADS52*, *Thiamine thiazole synthase 1* (*Thi1*), and *Histone deacetylase 110* (*hda110*), whose paralogs participate in multiple plant developmental pathways[41,42].

## Key TBN QTGs were subject to strong selection during maize breeding

Tassel branches were the subject of convergent selection in maize inbred lines from the United States and China during modern breeding[11,43–45]. To further understand the selection scenario of TBN during modern maize breeding, we calculated the nucleotide diversity of the seven verified candidate genes across teosinte entries, landraces and maize inbred lines using data from maize HapMap v3[46]. We detected dramatic nucleotide changes over the length of *qTBN8-2* (*lrs1*) (Fig. 5a). For this gene, nucleotide diversity of maize was significantly lower than that of teosinte and landraces, and that of landraces was apparently lower than that of teosinte, although this difference did not reach significance. This result indicated that *lrs1* might have been subjected to selection during maize improvement.

Further, we investigated the allele frequency at *lrs1* using data from maize HapMap v3[46]. We observed a gradual increase in the frequency of alleles causing less TBN over the course of maize

domestication and improvement (Fig. 5b), with a rise from 24% (teosinte) to 55% (landraces) and up to 66% (maize). This result is consistent with a selection signature at *lrs1* during maize improvement.

To further understand the genetic consequence of selection at *lrs1* during maize improvement, we conducted an association analysis and allele frequency analysis using published resequencing data, which included 350 elite maize inbred lines from China and the United States[11]. Association analysis of *lrs1* showed that four single nucleotide polymorphisms (SNPs) within this locus are significantly associated with TBN, suggesting that *lrs1* perhaps contributes to the phenotypic variation of TBN across elite maize inbred lines (Fig. 5c). By investigating allele frequencies of *lrs1* in the 350 elite maize inbred lines, we noticed that the frequency of the allele conferring less TBN was high in both China and the United States and rose slightly during modern maize breeding (Fig. 5d). Moreover, the less TBN allele contributes to the lower TBN values measured in inbreds from both China and the United States (Fig. 5e). We conclude that *qTBN8-2* (*lrs1*) underlies some of the phenotyping variation of TBN and was subjected to strong selection in maize.

## Significant selection of TBN pathways between female and male heterotic groups during modern maize hybrid breeding

A recent study uncovered 4804 candidate genes subject to selection in two heterotic groups during modern maize hybrid breeding. These selection genes were categorized into three groups: specific to male heterotic groups (MHGs), specific to female heterotic groups (FHGs), and co-selected in both MHGs and FHGs[45]. We compared the 4278

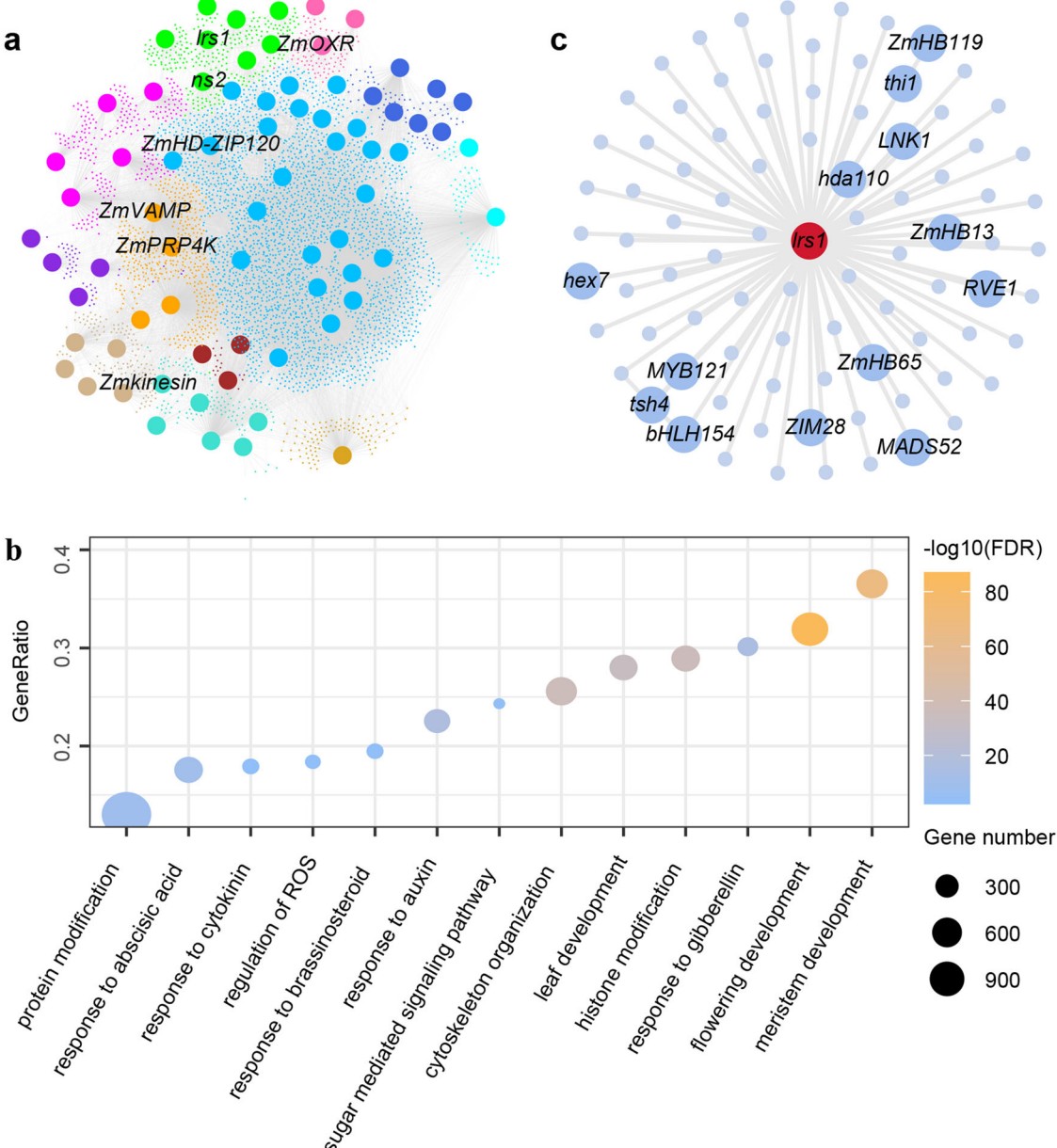

**Fig. 4 | Assembly of a comprehensive molecular network underlying TBN in maize. a** Constructed network of TBN in maize. The seven genes identified in this study are indicated. Different colors indicated different biological pathways as follows: Hotpink, abscisic acid and reactive oxygen species; Royalblue, auxin; Lime, boundary; Brown, brassinosteroid; Cyan, cytokinin; Turquoise, cytoskeleton and cellulose; Magenta, flowering; Tan, gibberellin; Goldenrod, histone modification; Deepskyblue, meristem maintenance and determinacy; Orange, protein modification and transport; Blueviolet, sugar and nutrition. **b** Significant enrichment of 4278 network genes in multiple biological pathways. **c** Subnetwork of *qTBN8-2* (*lrs1*).

genes from our constructed TBN network to these 4804 selection genes and determined that 40 (0.94%), 203 (4.75%), and 344 (8.04%) of our TBN network genes are FHGs, MHGs, and co-selected genes, respectively (Supplementary Fig. 6). GO enrichment analysis showed that 344 co-selected genes are significantly enriched in 470 terms and 203 MHGs were significantly enriched in 124 terms (Supplementary Data 18 and 19), while 40 FHGs showed no significant enrichment. These results indicate that MHGs and co-selected genes predominantly function in the convergent selection of TBN at the pathway level during modern maize hybrid breeding.

To further explore the convergent selection of TBN trait at the pathway level between the FHGs and MHGs, we compared the 4278 genes in the TBN network to the 1017 selection genes containing nonsynonymous SNPs whose allele frequencies exhibited co-

directional or anti-directional changes between FHGs and MHGs[45]. We determined that allele frequencies of 88 (2.06%) genes change co-directionally between FHGs and MHGs, while the allele frequencies of 66 (1.54%) genes changed convergently only in MHGs, and the allele frequencies of three (0.70%) genes changed anti-directionally between FHGs and MHGs (Fig. 6a). Enrichment analysis showed that 88 co-directional genes and 66 convergent genes exhibit a significant enrichment in our TBN network compared to a random sampling of equal gene number (Fig. 6b). The 88 co-directional genes and 66 convergent genes belonged to 10 biological pathways (Supplementary Fig. 7). About 65% of co-directional and convergent genes were associated with meristem maintenance and determinacy (Fig. 6c). GO enrichment analysis showed that 88 co-directional genes are enriched in 249 terms (Fig. 6d, Supplementary Data 20), which includes multiple

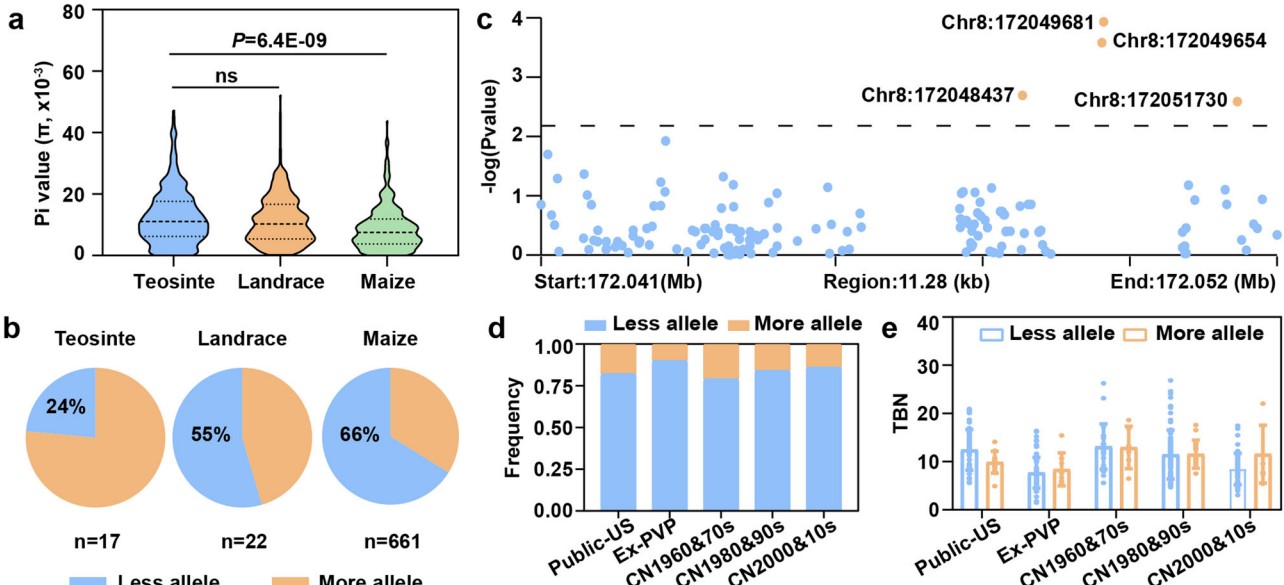

**Fig. 5 | Selection analysis at *qTBN8-2* (*lrs1*). a** Nucleotide diversity (pi value) across teosinte, landraces, and maize. *n* indicate 380, 387, 356 independent data points from left to right, respectively. ns, not significant. *P* values were determined by two-sided Student's *t* tests. ***\**P* < 0.001. Violin plots: median ± upper and lower quartiles. **b** Allele frequency of *lrs1* across teosinte, landraces and maize. Less allele, allele associated with less TBN; More allele, allele associated with more TBN. **c** Association signals over *lrs1* with TBN in a maize diversity panel. Four significant SNPs were uncovered. **d** Allele frequency of the SNP (Chr8: 172048437) in *lrs1* across different modern breeding populations. **e** Variation in TBN across different populations, sorted as a function of the lead association SNP. *n* indicate 57, 13, 75, 8, 22, 6, 76, 15, 44, 7 independent inbred lines from left to right, respectively. Data represent means ± s.d.

biological pathways such as meristem maintenance and determinacy, while 66 convergent genes were only enriched in two terms (Supplementary Data 21). Taken together, these results suggest that the significant selection of TBN occurred at multiple biological pathways between FHGs and MHGs during modern maize hybrid breeding, especially at the pathway of meristem maintenance and determinacy.

## Discussion

The fast and accurate fine mapping and cloning of functional genes underlying QTLs for agronomic traits is a critical component of crop improvement. Here, we developed a method, QTG-Miner, for the large-scale and efficient fine mapping and cloning of QTGs in maize. Additionally, using QTG-Miner, we rapidly dissected the causal genes behind seven TBN QTLs with both major and minor effects ranging from 4% to 14% in maize. Selection analysis showed that *qTBN8-2* (*lrs1*) contributed to the phenotypic variation of TBN in maize. Furthermore, we constructed a comprehensive molecular regulatory network underlying TBN in maize, which helped us uncover significant co-directional selection signatures enriched at multiple biological pathways between FHGs and MHGs during modern maize hybrid breeding. QTG-Miner is a high sufficient approach for systematically dissecting the genetic and molecular mechanisms underlying important agronomic traits in crops.

Most agronomic traits such as TBN are quantitative and controlled by many QTLs with major or minor effects. Conventional methods employ time-consuming and labor-intensive procedures, and only relying on conventional methods for QTGs cloning is inefficient and challenging, especially when dissecting such QTLs with minor effects or represented by rare alleles[1,8]. Combining QTG-Miner with conventional methods for QTGs cloning displayed obvious advantages for QTGs cloning. First, by integrating QTG-Miner with conventional methods, both major- and minor-effect QTLs could be rapidly cloned. A large number of recombinants enabled it possible that the selection of materials segregating for a single QTL alleviated potential interference of unlinked and background QTLs to a great extent, which helped ensure the accuracy of the identified DEGs and

variants. Second, with the help of various populations constructed using conventional methods, QTG-Miner can achieve functional gene cloning rapidly and efficiently when the genetic background of the population is properly controlled. Population types can be biparental or multiparental, including selfing and backcross populations, derived RILs, backcross inbred lines, residual heterozygous lines, nested association mapping populations, or multiparent advanced generation intercross populations. Third, integrating QTG-Miner with conventional methods could achieve large-scale gene cloning in a rapid and efficient manner. With the help of QTG-Miner, iterative and tedious fine mapping could be alleviated, and batch cloning the functional genes could be achieved at a time. Therefore, integrating QTG-Miner with conventional methods is a powerful strategy for accelerating QTGs cloning.

Meanwhile, several limitations or shortcomings of QTG-Miner also should be considered and overcome. Similar to conventional QTL mapping, it is a big challenge for QTG-Miner to discriminate the co-localized independent QTGs and clone them all in one. One alternative to overcome this difficulty is to nominate several (2 or 3) candidate genes and conduct phenotyping validation. In addition, QTG-Miner could not identify the causal genetic polymorphisms for QTGs directly. Several other methods could aid to dissect the causal genetic polymorphisms. Full-length resequencing of functional genes between two parents, candidate gene association analysis and haplotype analysis could help to identify the functional variants, as well as various molecular and genetic experiments.

TBN is an important agronomic trait that affects plant architecture and grain yield in maize. Following the isolation of relevant mutants, dozens of genes affecting TBN have been cloned and functionally dissected, which uncovered several genetic pathways such as meristem maintenance and determinacy[22,24,47,48], phytohormones[49-52], or sugar and nutrition[53]. However, only a few quantitative TBN genes have been verified, which limits our understanding of TBN genetic architecture and how to harness this trait in crop breeding[30,31].

Using QTG-Miner, we rapidly cloned and verified seven TBN QTGs, which greatly expands our understanding of tassel branch

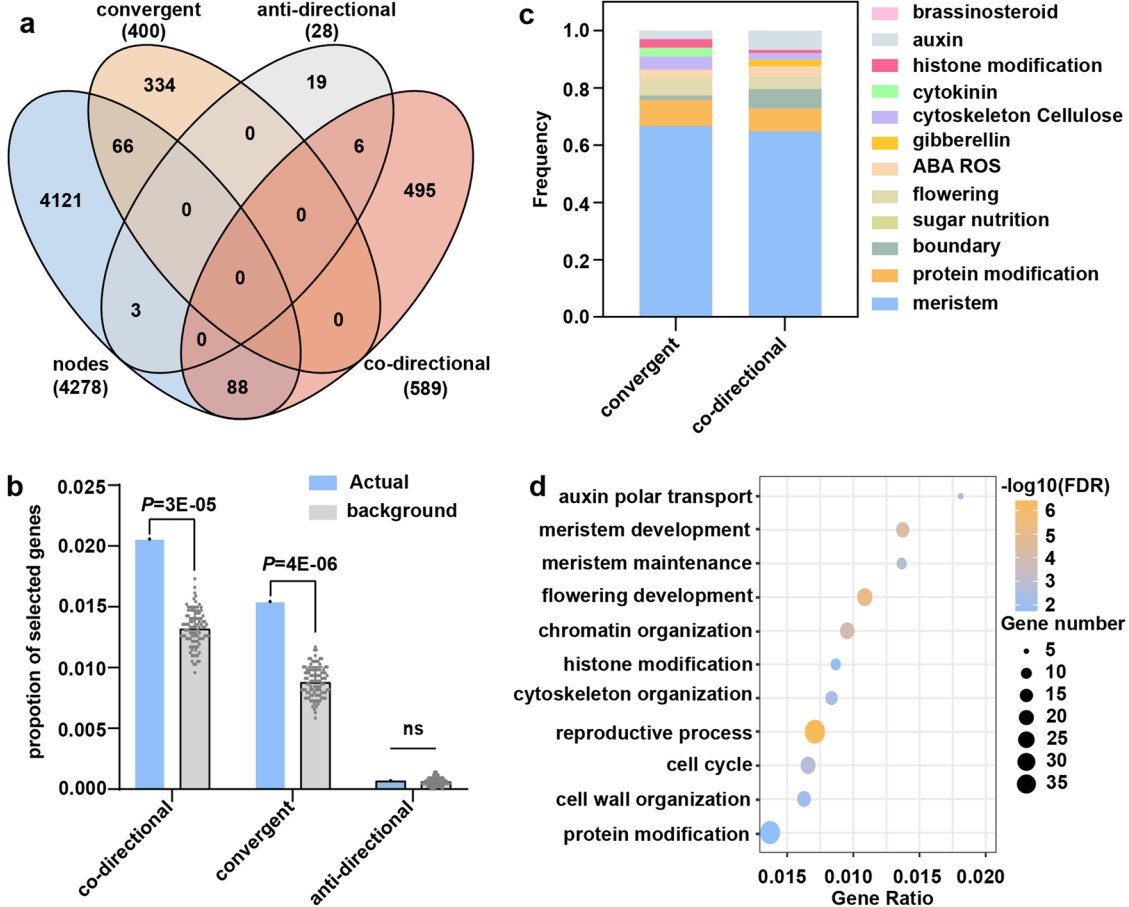

Fig. 6 | Co-directional selection on TBN pathways between FHGs and MHGs during modern maize hybrid breeding. a Venn diagram showing the extent of overlap among node genes of the GRN, convergent selected genes only in MHGs, anti-directional selected genes, and co-directional selection genes between MHGs and FHGs. b Significant enrichments of convergent selected genes only in MHGs and co-directional selection genes between MHGs and FHGs in the constructed GRN. n indicate 100 independent experiments among three type background genes. Data represent means ± s.d. P values were determined by Chi-squared test. c Frequency of convergent selected genes only in MHGs and co-directional selection genes between MHGs and FHGs among 12 biological pathways. d Significant enrichment of 88 co-directional selection genes in multiple biological pathways.

number trait in maize. For example, *qTBN4-1* encodes the TLD-domain containing nucleolar protein OXIDATION RESISTANCE (ZmOXR), whose ortholog in Arabidopsis, AtOXR2, improves photosynthesis efficiency, leading to higher plant growth and biomass production[54]. *qTBN4-2* encodes the WUSCHEL-related homeobox 3 protein, narrow sheath 2 (NS2). The mutant of *ns2* exhibits a brachytic plant phenotype with narrow sheath and shortened internodes[55,56]. Its putative ortholog in Arabidopsis is PRESSED FLOWER (PRS), which is required for lateral sepal development[57]. *qTBN8-2*, encodes a paralog of LIGULELESS2 (LG2), LRS1, which affects boundary identification and tassel branching in maize[58]. *qTBN8-3* encodes the CMGC_DYRK-PRP4 kinase, ZmPRP4K, whose putative ortholog in Arabidopsis AtPRP4KA functions in plant architecture and flower development including branching[59]. *qTBN10-2* encodes a vesicle-associated membrane protein ZmVAMP. In Arabidopsis, AtVAMP721/722 function in post-Golgi trafficking and are required for auxin-mediated development[60].

Multi-dimension omics data have been proved to exhibit great conveniences for systematically dissecting the genetic mechanisms behind important agronomic traits in model plants and crops[15–17,40,61–63]. In this study, we provided a approach for rapid and batch cloning of QTGs, which integrated the first-generation multi-omics network map and transcriptomic data obtained from RIL materials[37]. The first-generation multi-omics network map integrated multi-omics data including genome, transcriptome, translatome and proteome. Besides, the other types of omics data including

epigenomic and metabolomic data, also could aid the identification of functional genes[64,65]. These types of data could integrate into multi-omics network map and function as attributes in machine learning for aiding candidate gene mining.

By integrating the first-generation multi-omics network map, GRNs, and DEGs between mutants of the cloned TBN genes and their wild-type siblings, we constructed a comprehensive molecular network underlying TBN. Based on this TBN molecular network, we propose that ZmHD-ZIP120 might affect TBN via the meristem maintenance and determinacy pathway, while NS2 and LRS1 might affect TBN by identifying boundaries between organs. ZmOXR may be involved in ABA-mediated regulation of TBN, while ZmPRP4K and ZmVAMP might function in protein modification and transport, and ZmKinesin might function in cytoskeleton development. Our assembled TBN molecular network provides insights into the molecular mechanisms underlying TBN in maize.

TBN has been a major selection target for the observed decreasing tassel size during maize domestication and improvement[11,20,43–45]. So far, many TBN genes have been verified that have been the subject of selection during maize domestication or improvement[11,20,45]. Beyond the single gene level, our study uncovered the selection trajectory at the pathway level. Based on the assembled TBN network, we detected significant selection signatures of TBN at multiple biological pathways during modern maize improvement, which provides pathway targets for the systematic large-scale improvement of maize.

Although we tested and validated the procedure of QTG-Miner on TBN in maize, we expect that QTG-Miner can be extended to other traits and species. For example, QTG-Miner should be easily implemented on the model species Arabidopsis and rice (*Oryza sativa*) because of their small genome size, moderate genome complexity, abundant multi-omics data, and many known causal genes. However, QTG-Miner may be somewhat limited when implemented for non-model species. In such a scenario, it would be necessary to generate large-scale multi-omics data and assemble sets of positive genes prior to starting machine learning predictions. The rapid advance of next-generation sequencing technologies could help alleviate such challenges. Meanwhile, a series of studies have verified that many causal genes underlying important agricultural traits, for example plant height, flowering time, and tassel-related traits, are selection targets in evolution and domestication, and are somehow conserved across species[66–68]. Therefore, positive gene datasets can be obtained by identifying putative orthologs to known causal genes in model plant species. Thus, QTG-Miner has the potential to be widely applied to various species and traits.

## Methods

### Plant materials, growth conditions and phenotypic measurements

Twelve pairs of single QTL-segregating RIL materials were collected from 10 maize RIL populations[36]. Single QTL-segregating RIL materials and their parental inbred lines were selected and sown at the Huazhong Agricultural University experimental station (Sanya, Hainan province). Young leaf tissues of parental inbred lines were harvested for genomic DNA extraction. The immature tassel tissues (2–4 mm) of RIL materials with only one segregating QTL were collected for total RNA extraction (each replicate consisting of immature tassel tissues from six independent individuals). All samples were immediately frozen in liquid nitrogen and subsequently stored at –80 °C. EMS-mutagenized materials for 11 candidate QTGs were obtained from the maize EMS mutant library (https://elabcaas.cn/memd/public/index.html#/) and were planted in two locations for phenotypic investigation (Sanya, Hainan province and Zibo, Shandong province). Phenotypic investigation of *ZmKinesin* and *ZmHD-ZIP120* were conducted in Huazhong Agricultural University experimental station (Wuhan, Hubei province; Sanya, Hainan province). For the gene expression profile analysis of EMS-mutagenized materials and their wild-type sibling counterparts, populations were planted in Huazhong Agricultural University experimental station (Wuhan, Hubei province). Immature tassel tissues (from tassels 2–4 mm in length) were collected for total RNA extraction (two replicates per genotype, with each replicate comprising immature tassel tissues from six independent plants). TBN, taken as primary tassel branch number here, was measured after the genotyping was concluded. For field trials, each mutant plot was planted in replicate with a neighboring wild-type control plot. Two replicates were used for these phenotyping trials, with a row length and row space of 2.5 m and 60 cm, respectively. The space between plants in the same row was 25 cm.

### Genomic DNA and total RNA extraction and RNA-seq analysis

All DNA samples referred here were extracted using the CTAB method with minor modifications[69]. Total RNA from immature tassel tissues was extracted using a Direct-zol RNA Miniprep Kit (ZYMO RESEARCH) according to the manufacturer's instructions. mRNAs of RIL materials were subjected to sequencing on a Novaseq 6000 instrument (Illumina) as 150-bp paired-end reads. mRNAs of EMS-mutagenized materials were subjected to sequencing on a BGISEQ-500 instrument (Illumina) as 150-bp paired-end reads. For each biological replicate of RIL material, we obtained about 6 Gb clean data (40 million reads). For each biological replicate of EMS-mutagenized material, we obtained at least 20 Gb clean data (150 million reads) (Supplementary Data 8).

For RNA-seq analysis, clean reads were mapped to the maize reference genome (B73 RefGen_v4.36) using Hisat2 software (version 2.2.0)[70]. Gene expression levels were calculated using the fragments per kilobase of exon model per million mapped fragments (FPKM) method using cuffdiff (version 2.2.1)[71]. A GTF file containing the genomic coordinates of exons, and coding sequences of nuclear genes for maize (B73 RefGen_v4.36) was used to guide the annotation. Expression differences were considered statistically significant if $q < 0.05$, where $q$ is the $P$ value adjusted for multiple tests to minimize the false discovery rate (FDR). Genetic variants were called using GATK (version 3.7)[72] with default parameters; their effects on genes were annotated and predicted using SnpEff (version 4.3t)[73]. For sequence variants, based on their predicted effect on gene expression or protein structure, 13 types of major-effect variants were selected (conservative in-frame deletion, conservative in-frame insertion, disruptive in-frame deletion, disruptive in-frame insertion, frameshift, splice acceptor, splice donor, splice region, lost start codon, gained stop codon, lost stop codon, 5_prime_UTR_variant and 3_prime_UTR_variant). DEGs and genes with major-effect variants were considered as proportional weights during candidate gene mining.

### ML and SD algorithms on the multi-omics network map in QTG-Miner

The ML algorithms implemented in this study followed the method of Han et al. with slight modifications (https://github.com/hanlinqian/IntegrativeNetworkMap/tree/INM/Section5). A total of 32 TBN genes were compiled from the literature into the positive training dataset, and 55 non-TBN genes were curated from the classical maize gene database (https://genomevolution.org/wiki/index.php/Classical_Maize_Genes) and used as the negative training dataset (Supplementary Data 11). Log₂-normalized transcripts per million (TPM) values from RNA-seq data covering 31 tissues/stages and 21 tissues/stages from translatome data were used as co-expression and co-translation attributes, respectively. A total of 91 network attributes were collected from the resulting interactome, consisting of node information: eccentricity, closeness, degree, eigen centrality (four attributes) (https://github.com/hanlinqian/IntegrativeNetworkMap/blob/INM/Section1/code/NetInfo.r); and shortest distance to training positive and negative genes (87 attributes) (https://github.com/hanlinqian/IntegrativeNetworkMap/blob/INM/Section1/code/NetInfo.r). A total of 143 attributes were collated for subsequent predictions.

Five classical ML algorithms – bagging, XGBoost, LR, NeuralNet and SVM – were used to train the prediction model, and only NeuralNet algorithm was used to prioritize candidate TBN-related genes. To improve the stability of model evaluation, the data were randomly divided into two groups, with 80% as known data and the remaining 20% as unknown data; this analysis was repeated 20 times to calculate the AUC values to evaluate each model. For each predicted gene, a probability value was obtained. DEGs and sequencing variants identified from RNA-seq data derived from RIL materials would function as proportional weight as follows:

$$P_w = \frac{(P_n + P_d + P_v)}{3} \tag{1}$$

$P_w$ is the weighted probability of a certain gene. $P_n$ is the probability obtained from the NeuralNet algorithm, ranging from 0 to 1. $P_d$ and $P_v$ are the weights of the DEG and variants, respectively. If a gene was identified as a DEG, its $P_d$ value equals 1, and otherwise is equal to 0. If a gene possessed sequencing variants as referred above among both coding region and UTR region, its $P_v$ value equals 1, while a gene possessed sequencing variants among either coding region or UTR region, its $P_v$ value equals 0.5, and otherwise is equal to 0.

Total SD values in this dataset were stored in the GEO (Gene Expression Omnibus, https://www.ncbi.nlm.nih.gov/geo/) under

accession number GSE199932 as GSE199932_sd-slimio-highconf.txt.gz. The distance value of each gene to 57 known TBN-related genes was calculated, with the weight of one-layer distance of 1 and the second-layer distance being 0.01, and the sum being the final value ($SD_g$):

$$SD_g = \sum_{i=1}^{57} \begin{cases} SD_g + 1, SD_{gi} = 1 \\ SD_g + 0.01, SD_{gi} = 2 \end{cases} \quad (2)$$

For each predicted gene, the corresponding $SD_g$ value was obtained. DEGs and genic sequence variants identified from RNA-seq data derived from RIL materials would function as proportional weight as follows:

$$SD_w = \frac{(SD_g + SD_d + SD_v)}{3} \quad (3)$$

$SD_w$ is the weighted SD value of a certain gene. $SD_g$ is as described above. $SD_d$ and $SD_v$ are weights for the DEG and variants, respectively. If a gene was identified as a DEG, its $SD_d$ value equals to largest $SD_g$ among target QTL region, and otherwise equals 0. If a gene possessed sequence variants as referred above among both coding region and UTR region, its $SD_v$ value is equal to the largest $SD_g$ among target QTL region, while a gene possessed sequencing variants among either coding region or UTR region, its $SD_v$ value equals to half of the largest $SD_g$, and otherwise equals 0.

### Candidate gene mining and validation of causal sites
Candidate genes were prioritized based on the prediction results of the ML and SD algorithms. Candidate genes were selected by integrating gene functional annotations and literature on paralogs from Arabidopsis and rice. Genes with differential expression and sequence variants as referred to above were amplified and sequenced by Shanghai Sangon Biotech.

### Knockout of ZmKinesin and ZmHD-ZIP120 by CRISPR/cas9 systems
The CRISPR/Cas9 constructs for *ZmKinesin* and *ZmHD-ZIP120* were generated. The specific guide-RNAs designed for *ZmKinesin* and *ZmHD-ZIP120* were incorporated into the pCPB-*ZmUbi-hspCas9* vector, respectively (Fig. 3 and Supplementary Data 13)[74]. All constructs were introduced into the Agrobacterium strain EHA105 and transformed into the immature embryo of the maize inbred line KN5585 through Agrobacterium-mediated transformation. CRISPR/Cas9 knockout experiments of *ZmKinesin* and *ZmHD-ZIP120* were conducted by Wimi Biotechnology Co., Ltd. (Changzhou, China).

The target regions of *ZmKinesin* and *ZmHD-ZIP120* were amplified from KN5585 and corresponding transgenic lines and sequenced to identify the mutations. For *ZmKinesin*, we obtained two independent homozygous knockout lines named *Zmkinesin-KO#1* and *Zmkinesin-KO#2*. For *ZmHD-ZIP120*, we obtained one independent homozygous knockout line *Zmhd-zip120-KO#1* (Fig. 3 and Supplementary Data 13). The relevant primers used are listed in Supplementary Table 2.

### Nucleotide diversity, candidate gene association mapping and allele frequency analysis
To investigate the selection signature of seven genes identified in this study during maize domestication or improvement, the maize HapMap v.3 was downloaded (http://pan.baidu.com/s/1eRNGtxw)[46]. The variants among teosinte, landraces and maize were extracted and used to calculate nucleotide diversity (Pi value) and allele frequencies. Pi values were calculated with VCFtools (version 0.1.16) with "--max-missing 0.1 --mac 0.05 --recode --recode-INFO-all" parameters after removing SNPs with missing values greater than 10% or minor allele frequency (MAF) < 5%.

To further investigate the selection signature of *qTBN8-2* (*lrs1*) during modern maize breeding, a candidate gene association mapping and allele frequency analysis were conducted in a modern breeding population consisting of diverse genetic inbred lines. Candidate gene association mapping was conducted with 350 maize inbred lines, which are widely used in modern maize breeding programs in both China and United States[11]. Variants from upstream of the start codon (-2000 bp) to downstream of the stop codon (-2000 bp) in each candidate gene were extracted and analyzed. SNPs with MAF > 5% were used in the analysis. The Blink model was selected for the detection of SNPs significantly associated with TBN using the GAPIT program[75]. Allele frequencies of *qTBN8-2* (*lrs1*) were calculated across 350 maize inbred lines widely used in both China and United States.

### GRN construction
The multi-omics GRNs were constructed based on transcriptome and translatome datasets across maize development of the maize reference inbred line B73[40] and deposited at http://zeasystemsbio.hzau.edu.cn/dataset.html.

### Network construction of TBN
Construction of the molecular network of TBN was performed based on four different types of data: interaction edges, TF-centered GRNs, DEGs from six EMS-mutagenized materials, and the regulatome identified by tsCUT&Tag of ZmHD-ZIP120. Interaction edges of 72 TBN genes were obtained from the website http://minteractome.ncpgr.cn/index.php. For TF-centered GRNs, all regulatory pairs of 72 TBN genes inferred from both the transcriptional and translational levels were extracted, and regulatory pairs whose weights were larger than 0.02 were selected for network construction. DEGs of six cloned TBN genes between EMS-mutagenized materials and their wild-type sibling counterparts were identified using the methods described above and considered as interaction edges. Potential target genes of ZmHD-ZIP120 were identified using tsCUT&Tag[39]. High-confidence target genes of ZmHD-ZIP120 were identified from two independent biological replicates. Target genes of ZmHD-ZIP120 were further identified between the high-confidence target genes above and potential targets identified by the multi-omics GRN. Overlapping target genes were used as interaction edges during network construction. Modules in the network were clustered using the Markov cluster algorithm (https://micans.org/mcl/). Gephi (https://gephi.org/) was used for visualization and feature extraction from the networks.

### Selection signature analysis of TBN
Previously, 4804 genes were reported to be selected in at least two heterotic groups during modern hybrid maize breeding[45]. To further investigate their selection on TBN during modern maize breeding, a comparative analysis was conducted between these 4804 genes and the genes in the TBN network. MHG-specific selected genes, FHG-specific selected genes, and co-selected genes in both MHGs and FHGs were identified in the TBN network. Further, 4278 TBN network genes were compared to 1017 selection genes containing nonsynonymous SNPs and with an allele frequency exhibiting co-directional or anti-directional changes between FHGs and MHGs[45]. Genes changed co-directionally between FHGs and MHGs, changed convergently only in MHGs, and genes changed anti-directionally between FHGs and MHGs were identified. A GO enrichment analysis was then conducted.

### GO enrichment analysis
GO enrichment analysis was performed with agriGO (v.2.0) using the Singular Enrichment Analysis (SEA) method. GO terms were considered significantly enriched if the Bonferroni false discovery rate was below 0.05.

**Reporting summary**

Further information on research design is available in the Nature Portfolio Reporting Summary linked to this article.

## Data availability

Sequence data discussed in this article can be found in the MaizeGDB (https://www.maizegdb.org/) gene records under the following accession numbers: *qTBN3-1* (Zm00001d042795), *qTBN4-1* (Zm00001d052219), *qTBN4-2* (Zm00001d052598), *qTBN7-1* (Zm00001d020804), *qTBN8-2* (Zm00001d012295), *qTBN8-3* (Zm00001d012452), *qTBN10-2* (Zm00001d025939). Maize reference genome (B73 RefGen_v4.36) was downloaded using command in linux system [wget ftp://ftp.ensemblgenomes.org/pub/release-36/plants/fasta/zea_mays/dna/Zea_mays.AGPv4.dna.toplevel.fa.gz]. The raw sequencing data generated in this paper have been deposited in the Genome Sequence Archive[76] in National Genomics Data Center[77], China National Center for Bioinformation/Beijing Institute of Genomics, Chinese Academy of Sciences under accession CRA009203. Source data are provided with this paper.

## Code availability

Machine learning codes for QTG-Miner are available at GitHub [https://github.com/hanlinqian/IntegrativeNetworkMap/tree/INM/Section5]. Shortest distance codes for QTG-Miner are available at GitHub [https://github.com/hanlinqian/TBNnetwork]. Codes are also available at *Zenodo* (https://doi.org/10.5281/zenodo.8246635)[78].

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

## Acknowledgements

This research was supported by the National Natural Science Foundation of China (31922068 to L.L.), Hainan Yazhou Bay Seed Lab (B21HJ8102 to L.L.), the major program of Hubei Hongshan Laboratory (2021hszd008 to L.L.), and the Outstanding Youth Team Cultivation Project of Center Universities (2662023PY007 to L.L.). We thank the high-performance computing platform at the National Key Laboratory of Crop Genetic Improvement in Huazhong Agricultural University. We also thank Dr. Chunyi Zhang from the Chinese Academy of Agricultural Sciences for the construction of EMS mutant library.

## Author contributions

L.L. designed and supervised this study. X.W., Y.Q., J.L. and J.X.L. collected raw data. X.L contributed EMS mutants. X.W., J.L., L.H., C.L., X.S., X.M., Z.Luo, W.Z., Z.L. and T.L. performed the data analysis. L.L., X.W. and J.L. wrote the manuscript. H.L. provided constructive comments and suggestions for manuscript improvement. All authors have read and approved the manuscript.

## Competing interests

The authors declare no competing interests.
