## [Peer Review File · Nature Communications]

QTG-Miner aids rapid dissection of the genetic base of tassel branch number in maizeReviewers' Comments:

Reviewer #1:

Remarks to the Author:

The manuscript by Wang et al reports a novel multi-omics based approach to determine causal genes underlying QTL. Their method, QTL-miner, is separated 3 components where QTL are mapped, transcriptomes of individuals from the mapping population that have divergent genotypes in the QTL region and largely genetically similar outside the QTL are generated, and integration of multi-omics data to rank likelihood a given gene is responsible for the QTL. They use this approach to identify candidate genes underlying QTL for tassel branch number and go on to verify a high proportion of the candidates using EMS and crispr/CAS mutants. This approach expands and improves on previous network-based approaches to identify causal genes underlying QTL. Overall this is a well written manuscript with exciting findings that are well supported by the data and is broadly applicable to quantitative geneticists studying a wide-range of species.

Here are some very minor comments to consider:

- 1) In the abstract qTBN8-2 is stated to be a gene. My understanding is that qTBN8-2 is the QTL and *Lrs1* is the gene.
- 2) The allele naming is a bit confusing. For example, I think the EMS allele for ZmKinesin should be designated; possibly as *zmkinesin-1*. Also, in Figure 3g, is the 3rd sequence supposed to be *zmkinesin-2*?
- 3) I am not exactly clear on what data is in Supplemental Table 8. Perhaps an expanded description of the analysis leading to the data output in S8 would be beneficial.
- 4) What do the colors represent in Figure 5A?

Signed,
Justin Walley

Reviewer #2:

Remarks to the Author:

The authors have developed, tested and validated a new -omics data-based approach (QTG-Miner) for rapid and efficient large-scale fine-mapping and cloning of QTGs in maize. By using this method, the authors have cloned 7 QTGs for tassel branch numbers. The development of QTG-Miner and the efficient cloning of QTGs in maize are noteworthy results. Despite several limitations this approach can be potentially applied to other model and non-model species and it could greatly assist the genetic improvement and reduce the breeding time of several crops.

Several times throughout the manuscript the authors state that they developed a multi-omics data-based approach. In order to use this term they need to integrate data of at least 3 omics approaches. To my understanding they have used transcriptomic and genomic data. Nowadays, proteomic, epigenomic and metabolomic data is generated for several crop species. It would be very important and useful if these types of omics data can be also integrated in QTG-miner. Would this be possible with QTG-miner? To my opinion this needs to be discussed.

Moreover it is very common when studying traits with complex genetic architecture, the identification of a major quantitative trait gene hides several independent QTGs in the same region. How sensitive is QTG-Miner to dissect all these co-localised independent QTGs?

Minor points

Figure 1: please correct QTL-paried to QTL-paired

Figure 2: please explain in the legend the abbreviation of ChIA-PET as you do for PPI

Page 10, line 227: please change "this" with "qTBN3-1" because it is not easy for the reader to understand what "this" corresponds to.

lines 251: authors conclude that they have validated a high proportion of candidate genes. That is 58,3% (line 285). The other QTGs were not validated? Why did they choose these 7? Is there any evidence or proportion of false positive in their results?

lines 556-557: the depth of sequencing is not mentioned. What was the amount of sequencing data (Gigabases? or Million of reads?)

Reviewer #3:

Remarks to the Author:

The work of Wang et al. proposed a rapid strategy for mapping and cloning of QTLs, which includes initial QTL mapping, sequencing single QTL segregating material, and candidate gene mining by QTG-Miner. Specifically, they used two algorithms (shortest distance and machine learning) to rank the candidate genes and then screened EMS mutants or created CRISPR/Cas9 editing materials to validate their selected candidates for tassel branch number (TBN) in maize. They then constructed a molecular network for TBN based on this work and their previous work of interactome. They further evaluated the co-directional selection signatures of TBN network between female and male heterotic groups during modern maize hybrid breeding by using a recently published dataset. Overall, this work is interesting and has large amount of data, the contents and figures are well presented, and the methods are reasonable. Nonetheless there are some important points which the authors should consider to improve the analyses and presentation of the findings. Main concerns are the general application of QTG-Miner and the solidness of functional validation of QTLs.

Major points:

The highlight of this study would be that the authors used two algorithms in QTG-Miner - shortest distance (SD) and machine learning (ML) - for speeding up candidate gene mining. Based on the results, the algorithms SD and ML might prioritize candidate genes for some case (qTBN3-1 is a good example with 1st ranking in both SDw and Pw), however, most validated candidates don't seem to have good rankings of SDw and Pw (see SI Fig 1). The fact is that "candidate genes were selected by integrating gene functional annotations and literature on parlors from Arabidopsis and rice" according to methods described in L622-624, which just follows a common path for considering or guessing candidate genes under a given QTL region. The two type of rankings for most validated candidate genes appear to have minor advantages here. Therefore, the general application of QTG-Miner would be rather limited.

The authors compiled a list of 57 known functional genes affecting TBN as positive genes and 63 negative genes with no evidence of being connected to TBN, which has been a core dataset used in QTG-Miner for candidate gene mining. However, no functional annotation and citation was found for any of these genes in SI Table 4. More importantly, what were the standards to define TBN or non-TBN genes? Convincing evidences are needed to claim the use of these genes as proper references.

While calculating SDw and Pw, the authors added DEGs and sequence variants as proportional weights. Have you considered using the fold changes of DEGs rather than 0 vs. 1 as weights? That might assign reasonable weights for candidate genes in my opinion. For parental sequence variants, only those ones in genic region were used. Why not consider regulatory variation? A recent review paper of natural variation in crops shows that regulatory variation accounts for a majority of causal genetic polymorphisms for QTLs cloned in maize (Liang et al. Annu Rev Plant Biol 2021, 72:357-385). The authors actually showed 3'UTR variant (qTBN7-1, Fig 3M) and 3-bp InDel in the promoter (qTBN4-

2, SI Fig 1D) for two genes, but these variants were not used in QTG-Miner. Given the importance of regulatory variants (intergenic, promoters, UTRs, etc.), I would suggest the authors include them in QTG-Miner, too. And rather than use 0 vs. 1 as weights for sequence variants, maybe you could try 0, 1, 2 (0 - no variants, 1 - either coding or regulatory variants, 2 - both coding and regulatory variants) as weights.

It was good that the authors were able to obtain EMS mutants or CRISPR/Cas9 lines to perform functional validation of candidate genes. However, although the validated genes through mutants or edited-lines did affect TBN, it doesn't necessarily mean they are the actual functional genes. They could also be genes of pleiotropy. A pleiotropic gene might be mediated by distinct cis-regulatory variants. Have you investigated other typical agronomic traits for these genes? Such supporting data would be more convincing to claim they are actual TBN genes.

The authors used a whole section (Line 281-303) to stress the robustness of QTG-Miner. However, the evidences are not strong enough to claim the robustness. First, as stated above, the validated genes may not be the actual TBN genes. Second, except for one QTL (qTBN3-1) which has three mutation alleles (one EMS mutant and two Cas9-edited lines), the other six validated genes all have only one mutation allele, which is generally not enough (at least two mutation alleles or even overexpression alleles should be used) for functional validation. Third, it is important to evaluate traits of targeted genes across multiple years and multiple environments. From the TBN phenotypes of WT and mutants in Fig 3 and SI Fig 1, it looks like at least four QTLs (qTBN3-1, qTBN4-2, qTBN8-3, and qTBN10-2) have genotype-by-environment interaction effects. Mutants at qTBN8-3 and qTBN10-2 didn't even show significant TBN effects in one of the environments, and TBN was only evaluated at one environment one year for Cas9-edited lines. Therefore, I suggest the authors weaken the description of this section, especially avoid using percentages as the denominators are so small (7/12, 5/8, 2/4). Correspondingly, Figure 4 is better as a supplemental figure rather than a main figure.

Compared to traditional positional cloning, one weakness of the strategy proposed in this study is that it can't identify the causal genetic polymorphisms for QTLs. Knowing the causal variants of genes can largely help us understand the evolutionary path for TBN during maize domestication and improvement. The authors should include this important aspect in the discussion.

Other points:

Figure 3 - The sequence variant of parents (3D and 3M) is not information for the figure (the phenotypes are from mutants or gene-edited lines rather than parents), so it is better to move them to supplemental. I can barely see the difference of TBN in 3E. Do you have a better tassel picture? The picture of 3H is not representative as compared to the phenotypic means from 3I. There is about 6 and 8 TBN for WT and Cas9-edited lines in 3I, while I can only see 4 and 6 in 3H. Please indicate the genotypes in all tassel pictures and the sample sizes for all statistical tests in Fig 3 and SI Fig 1. The legends are redundant for the two examples.

How did the authors define the QTL region for candidate genes? I assume some level of support interval was used, but it might be not true. I'm sure the homozygous non-recombinants at target QTL were used for RNA-seq and sequencing. Such information is unclear until you specify them somewhere in the text.

All of QTL figures shown in the manuscript (Fig 3, SI Fig 1) seem to be schematic diagram. Real QTL mapping diagram won't have such smooth LOD curve. Please use the actual LOD curve mapped from the corresponding bi-parental population.

Figure 2 - In 2E and 2F, red and blue solid dot indicate positive and negative candidate genes according to the legends. I'm confused with the meanings of positive and negative. Why are red and blue alternatively distributed in QTLs?

SI Table 3 - It is surprising that the overlap of DEGs and genes with sequence variants is not very high. Among the seven validated candidate genes, four genes have sequence variants, one has DE, one has both, and one has neither. Can you explain this? Are there differential expression for these genes in WT vs. mutants or CRISPR/Cas9 lines?

Figure 1 - What is the scale for the 2-4mm tassel?

SI Table 2 - The sample name of Zm00001d012295 was incorrect. It should be 2295mu, 2295wt. By the way, what is the genetic background of the mutants? They also have much higher percentage of mapped reads than RILs.

SI Table 4 - Why is there a subset of genes not available for machine learning?

Line 33: The full names for *Irs1* should be given here for the first appearance.

Line 131: To show the genetic background at each TBN QTL, can you provide a supplemental graph with genotypic information of ten chromosomes for RILs used for each QTL and indicate the target QTL in the graph?

Line 222 - There are no CRISPR/Cas9 materials in SI Table 6.

Line 240 - "resulting in" is a strong word. Change the words as there is no experiment done in vivo or in vitro to verify that the 8-bp InDel is the actual variant affecting the transcript level.

Line 311 - SI Table 8 only has source and target for network. Where is the information of these 1,012 DEGs, such as transcript levels in WT and mutants, fold change, q value, etc.?

Line 530 - Which developmental stage was for the 2-4mm immature tassel? Does the RIL pair for each QTL have difference?

Line 538 - Can you provide details regarding CRISPR/Cas9 experiments, such as gRNA, transformation method, etc.?

Line 549 - For RCBD, how many blocks (replicates) and how many treatments (genotypes) within each block? How did you correct TBN phenotype for the experimental design?

Line 556 - I believe you were sequencing mRNA rather than total RNA.

Line 561 - Did you use default parameters for Hisat2 and Cuffdiff? What is the number of mismatches and the maximum number of multi-mapped positions allowed? Did you filter low read count before performing DE test? There are many genes with 0 expression level in one genotype, but you really don't know if those genes are actually not expressed or just because the sequencing library size doesn't reach the cutoff for them to be detected.

Reviewer #4:

Remarks to the Author:

The manuscript has several key topics: 1. The development of QTG-Miner, a method/algorithm for rapid cloning of candidate genes, 2. The genetics and mechanistic understanding of tassel branch number (TBN) in maize, 3. the selection history of TBN.

The manuscript presents an impressive data set and relevant analyses to address the above mentioned topics. It is well written.

The objectives of the manuscript are well presented in the introduction. The main focus is on the development of QTG Miner. The authors claim that QTG Miner is a novel method to speed up gene cloning for quantitative traits and outperform conventional methods with respect to time and costs.

The authors present 12 QTL regions controlling TBN identified in an earlier study. In seven of them QTG Miner identified causal genes that could be validated in EMS or CRISPR mutants. In addition they present a molecular network underlying TBN and analyze the selection pathways of TBN in male and female heterotic maize groups.

The manuscript would have profited from a more focused presentation of either the QTG Miner OR the mechanistic insights on TBN. By trying to cover both topics it has not been possible to cover either of the topics with sufficient depth.

With respect to QTG Miner quite a few important points were not covered:

The procedure needs a lot of prior information, eg mapped QTL and known genes regulating the trait. In the last paragraph of the discussion this is briefly mentioned. The list of positive genes for TBN in maize is long and this is not the case for most quantitative traits and crops. The efficiency of the method should have been tested with a smaller number of positive genes to determine a threshold and report on the required prior information.

One of the cloned genes (TBN7) was not the top hit in the SD and ML analyses (no 3 and 5). The authors did not describe how they converged on the causal gene from the list the algorithms provided even though others had higher scores.

It was not reported how many genes the 12 QTL regions comprised, how many candidates per QTL region were selected and how many failures in the validation were observed (eg how many mutants were analyzed in total to validate the seven genes).

What happened in the 5 QTL regions where no causal gene could be identified?

It was not described how the results from the SD and ML analyses differed with respect to the success criterion. The introduction of a new method requires a more detailed coverage with respect to parameter settings, specificity and sensitivity of the method.

It would have been interesting to see if in some QTL regions the algorithm was more successful than in others.

How was the list of negative genes chosen for the ML algorithm, why different numbers of positive genes in SD and ML?

How was the integrated network map built? The methods section should be more detailed on the chosen parameters.

A more detailed presentation of the results of the DEG analysis would have been useful.

In the discussion the authors claim that cost and time can be saved with QTG Miner. That is only the case if a lot of preliminary work has been done on the trait as is the case for TBN. A more critical discussion on what it takes and how QTG Miner can be combined with classical methods generating recombinants would have been useful. The workflow in Figure 1 is not really that novel as it is quite similar to what is done in today's mapping projects without prior information on candidates.

It would have been interesting to learn if prior information from related species would have been

useful (eg include them in the list of positive genes)

With respect to selection analyses and molecular networks:

Are these sections essential to introduce QTG Miner?

Isn't it counter-intuitive that most of the TBN genes are co-directional in the male and female heterotic pools?

**Point-by-point response to the reviewers' comments**

We sincerely thank all reviewers for their constructive comments and suggestions.
These comments greatly help us to improve the manuscript. We have conducted all the
experiments as suggested and addressed all the concerns, and revised the manuscript
accordingly. Our responses to reviewers' comments are in blue below, as well as in the
revised manuscript.

**REVIEWER COMMENTS**

Reviewer #1 (Remarks to the Author):

The manuscript by Wang et al reports a novel multi-omics based approach to determine
causal genes underlying QTL. Their method, QTL-miner, is separated 3 components
where QTL are mapped, transcriptomes of individuals from the mapping population
that have divergent genotypes in the QTL region and largely genetically similar outside
the QTL are generated, and integration of multi-omics data to rank likelihood a given
gene is responsible for the QTL. They use this approach to identify candidate genes
underlying QTL for tassel branch number and go on to verify a high proportion of the
candidates using EMS and crispr/CAS mutants. This approach expands and improves
on previous network-based approaches to identify causal genes underlying QTL.
Overall this is a well written manuscript with exciting findings that are well supported
by the data and is broadly applicable to quantitative geneticists studying a wide-range
of species.

**Response:** Thank you so much for your positive comments.

Here are some very minor comments to consider:

1) In the abstract qTBN8-2 is stated to be a gene. My understanding is that qTBN8-2 is
the QTL and lrs1 is the gene.

**Response 1.1:** Thank you for pointing out this. It has been revised as you suggested.

2) The allele naming is a bit confusing. For example, I think the EMS allele for
ZmKinesin should be designated; possibly as *zmkinesin-1*. Also, in Figure 3g, is the
3rd sequence supposed to be *zmkinesin-2*?

**Response 1.2:** Thank you so much for pointing out this. We should have clarified this
clearly previously. Following your suggestion, EMS allele for ZmKinesin has been
designated as *Zmkinesin-EMS*. Meanwhile, two CRISPR/Cas9 edited materials have
been designated as *Zmkinesin-KO#1* and *Zmkinesin-KO#2*, respectively.

3) I am not exactly clear on what data is in Supplemental Table 8. Perhaps an expanded
description of the analysis leading to the data output in S8 would be beneficial.

**Response 1.3:** Data in Supplementary Table 8 (Supplementary Table 9 in revised
manuscript) consists of two parts: DEGs identified from EMS materials and
downstream target genes of ZmHD-ZIP120 identified by tsCUT&Tag. By integrating
these data as interaction pairs, we constructed the integrated TBN network. According
to your suggestions, we revised and added detailed description as follows:

“To construct the molecular network underlying TBN, we performed RNA-seq
analyses on the EMS mutants of six of the seven newly cloned TBN genes and their
wild-type sibling counterparts (Supplementary Table 3). Totally, we identified 1,349
DEGs, which are potentially involved in the TBN network (Supplementary Table 9).
Additionally, we conducted transient and simplified Cleavage Under Targets and
Tagmentation (tsCUT&Tag)³⁹ on the protein encoded by the newly cloned TBN gene,
the homeobox leucine zipper domain transcription factor ZmHD-ZIP120 (the causal
gene for *qTBN7-1*), integrated the multi-omics gene regulatory network constructed in
our previous study for the dissection of the regulome of TBN genes⁴⁰, and identified
957 potential target genes totally (Supplementary Figure 5, Supplementary Table 9).”
(Lines 297-307)

4) What do the colors represent in Figure 5A?

**Response 1.4:** Different colors in Figure 5A (Figure 4A in revised manuscript)
represent different network modules divided by biological pathways. We added the
detailed description in figure legends.

“Different colors indicated different biological pathways as follows: Hotpink,
abscisic acid and reactive oxygen species; Royalblue, auxin; Lime, boundary; Brown,
brassinosteroid; Cyan, cytokinin; Turquoise, cytoskeleton and cellulose; Magenta,
flowering; Tan, gibberellin; Goldenrod, histone modification; Deepskyblue, meristem
maintenance and determinacy; Orange, protein modification and transport; Blueviolet,
sugar and nutrition.” (Lines 332-336)

Signed,

Justin Walley

Reviewer #2 (Remarks to the Author):

The authors have developed, tested and validated a new -omics data-based approach
(QTG-Miner) for rapid and efficient large-scale fine-mapping and cloning of QTGs in
maize. By using this method, the authors have cloned 7 QTLs for tassel branch numbers.
The development of QTG-Miner and the efficient cloning of QTGs in maize are
noteworthy results. Despite several limitations this approach can be potentially applied
to other model and non-model species and it could greatly assist the genetic
improvement and reduce the breeding time of several crops.

**Response:** Really appreciate your evaluation and comments on our manuscript.

Several times throughout the manuscript the authors state that they developed a multi-
omics data-based approach. In order to use this term they need to integrate data of at
least 3 omics approaches. To my understanding they have used transcriptomic and
genomic data. Nowadays, proteomic, epigenomic and metabolomic data is generated

for several crop species. It would be very important and useful if these types of omics
data can be also integrated in QTG-miner. Would this be possible with QTG-miner? To
my opinion this needs to be discussed.

**Response 2.1:** Thank you so much for your suggestion on this. Here, QTG-Miner was
based on a multi-omics integrative network map of maize showed in Figure 2A, which
integrates multi-omics data (ChIA-PET, Co-expression, Co-translation and PPI)³⁷.
These multi-omics data, as well as transcriptomic data obtained from RIL materials,
were used in machine learning for candidate gene mining. Yes, the other types of omics
data including epigenomic and metabolomic data, could indeed function as attributes in
machine learning and shortest distance for candidate gene mining. Following your
suggestion, we added the detailed description in the part of Results as well as
Discussion.

“With the rapid development of various sequencing methods, an integrative multi-
omics network map was assembled (Figure 2A) and evidenced to accelerate the
dissection of biological pathways and predict gene function³⁷, which integrated multi-
omics data from ChIA-PET (chromatin interaction analysis by paired-end tag
sequencing), co-expression, co-translation and PPI (protein-protein interaction).”
(Lines 190-194)

“Multi-dimension omics data have been proved to exhibit great conveniences for
systematically dissecting the genetic mechanisms behind important agronomic traits in
model plants and crops^{15, 16, 17, 40, 61, 62, 63}. In this study, we provided a new approach for
rapid and batch cloning of QTGs, which integrated the first-generation multi-omics
network map and transcriptomic data obtained from RIL materials³⁷. The first-
generation multi-omics network map integrated multi-omics data including genome,
transcriptome, translome and proteome. Besides, the other types of omics data
including epigenomic and metabolomic data, also could aid the identification of new
genes^{64, 65}. These types of data could integrate into multi-omics network map and
function as attributes in machine learning for aiding candidate gene mining.” (Lines

493-502)

Moreover it is very common when studying traits with complex genetic architecture,
the identification of a major quantitative trait gene hides several independent QTLs in
the same region. How sensitive is QTG-Miner to dissect all these co-localised
independent QTGs?

**Response 2.2:** Yes, it is a big challenge for QTG-Miner to discriminate the co-localized
independent QTGs and clone them all in one. What QTG-Miner could do is to prioritize
candidate genes based on the scoring, and previous biological knowledge about the
genes would be a bonus for candidate gene selection and validation pretty like that fine-
mapping requires. To alleviate your concern, we added discussion in the revised
manuscript as follows:

“Meanwhile, several limitations or shortcomings of QTG-Miner also should be
considered and overcome. Similar to conventional QTL mapping, it is a big challenge
for QTG-Miner to discriminate the co-localized independent QTGs and clone them all
in one. One alternative to overcome this difficulty is to nominate several (2 or 3)
candidate genes and conducting phenotyping validation.” (Lines 461-465)

**Minor points**

Figure 1: please correct QTL-paried to QTL-paired

**Response 2.3:** We sincerely apologize for this typo. We corrected it in the revised
manuscript. Thank you so much.

Figure 2: please explain in the legend the abbreviation of ChIA-PET as you do for PPI

**Response 2.4:** Thank you so much for reminder on this. We have revised the manuscript
and added the explanation of ChIA-PET as follows: “ChIA-PET, chromatin interaction
analysis by paired-end tag sequencing”.

Page 10, line 227: please change "this" with "qTBN3-1" because it is not easy for the

reader to understand what "this" corresponds to.

**Response 2.5:** Thank you so much for your suggestion on this. We changed "this" with
"*qTBN3-1*" in our manuscript.

lines 251: authors conclude that they have validated a high proportion of candidate
genes. That is 58,3% (line 285). The other QTGs were not validated? Why did they
choose these 7? Is there any evidence or proportion of false positive in their results?

**Response 2.6:** Thanks for your questions on this. Using QTG-Miner, we rapidly
narrowed down and targeted candidate genes underlying 12 TBN QTLs. For each QTL,
we chose only one candidate gene with our prior biological knowledge and conducted
phenotypic verification by EMS-mutagenesis and CRISPR-edited mutation. Among 12
TBN QTLs, 7 QTLs can be verified. Other 5 candidate genes underlying corresponding
QTLs could not be validated by EMS-mutagenesis, which might be caused by
functional redundancy of homologous genes, wrong nominations of candidate genes,
or other factors.

lines 556-557: the depth of sequencing is not mentioned. What was the amount of
sequencing data (Gigabases? or Million of reads?)

**Response 2.7:** Thank you so much for pointing out this. We are so sorry that we did
not clarify it clearly previously. The detailed information of RNA-Seq data was
summarized in Supplementary Table 3. Meanwhile, we introduced detailed description
of RNA-Seq data in Methods as follows:

"For each biological replicate of RIL material, we obtained about 6 Gb clean data
(40 million reads). For each biological replicate of EMS-mutagenized material, we
obtained at least 20 Gb clean data (150 million reads) (Supplementary Table 3)." (Lines
570-573)

Reviewer #3 (Remarks to the Author):

The work of Wang et al. proposed a rapid strategy for mapping and cloning of QTLs,
which includes initial QTL mapping, sequencing single QTL segregating material, and
candidate gene mining by QTG-Miner. Specifically, they used two algorithms (shortest
distance and machine learning) to rank the candidate genes and then screened EMS
mutants or created CRISPR/Cas9 editing materials to validate their selected candidates
for tassel branch number (TBN) in maize. They then constructed a molecular network
for TBN based on this work and their previous work of interactome. They further
evaluated the co-directional selection signatures of TBN network between female and
male heterotic groups during modern maize hybrid breeding by using a recently
published dataset. Overall, this work is interesting and has large amount of data, the
contents and figures are well presented, and the methods are reasonable. Nonetheless
there are some important points which the authors should consider to improve the
analyses and presentation of the findings. Main concerns are the general application of
QTG-Miner and the solidness of functional validation of QTLs.

**Response:** Thank you for your constructive comments and valuable suggestions, which
greatly helped us to improve the manuscript.

Major points:

The highlight of this study would be that the authors used two algorithms in QTG-
Miner - shortest distance (SD) and machine learning (ML) - for speeding up candidate
gene mining. Based on the results, the algorithms SD and ML might prioritize candidate
genes for some case (qTBN3-1 is a good example with 1st ranking in both SDw and
Pw), however, most validated candidates don't seem to have good rankings of SDw and
Pw (see SI Fig 1). The fact is that "candidate genes were selected by integrating gene
functional annotations and literature on parlors from Arabidopsis and rice" according
to methods described in L622-624, which just follows a common path for considering
or guessing candidate genes under a given QTL region. The two type of rankings for
most validated candidate genes appear to have minor advantages here. Therefore, the
general application of QTG-Miner would be rather limited.

**Response 3.1:** Thank you so much for pointing out this. QTG-Miner was developed as
an alternative for accelerating the procedure of fine mapping. What QTG-Miner could
do is to prioritize candidate genes based on the scoring, and previous biological
knowledge about the genes would be a bonus for candidate gene selection and
validation as fine-mapping requires. Based on this, it is understandable that the casual
genes do not always have good ranks. Even fine-mapping or association mapping, the
causal genes are usually not located in the highest linked peak regions and need prior
knowledge for the nomination.

The authors compiled a list of 57 known functional genes affecting TBN as positive
genes and 63 negative genes with no evidence of being connected to TBN, which has
been a core dataset used in QTG-Miner for candidate gene mining. However, no
functional annotation and citation was found for any of these genes in SI Table 4. More
importantly, what were the standards to define TBN or non-TBN genes? Convincing
evidences are needed to claim the use of these genes as proper references.

**Response 3.2:** Thank you so much for pointing out this. Positive genes referred here
had been cloned and verified affecting TBN in maize by a series of previous studies.
Meanwhile, these positive genes perhaps exhibit pleiotropy and affect other agricultural
traits, for example other plant architecture and yield-related traits. Negative genes
referred here had never been reported affecting tassel-related traits, and were screened
according to functional annotations, expression profiles, including some classical maize
genes (https://genomeevolution.org/wiki/index.php/Classical_Maize_Genes), house-
keeping genes (for example actin-related genes) and genes of tissue-specific expression
not detected in tassel. According to your suggestions, we added the functional
annotations and citations of these genes in Supplementary Table 5 (in revised
manuscript).

While calculating SDw and Pw, the authors added DEGs and sequence variants as
proportional weights. Have you considered using the fold changes of DEGs rather than
0 vs. 1 as weights? That might assign reasonable weights for candidate genes in my

opinion. For parental sequence variants, only those ones in genic region were used.
Why not consider regulatory variation? A recent review paper of natural variation in
crops shows that regulatory variation accounts for a majority of causal genetic
polymorphisms for QTLs cloned in maize (Liang et al. Annu Rev Plant Biol 2021,
72:357-385). The authors actually showed 3'UTR variant (qTBN7-1, Fig 3M) and 3-
246 bp InDel in the promoter (qTBN4-2, SI Fig 1D) for two genes, but these variants were
247 not used in QTG-Miner. Given the importance of regulatory variants (intergenic,
promoters, UTRs, etc.), I would suggest the authors include them in QTG-Miner, too.
And rather than use 0 vs. 1 as weights for sequence variants, maybe you could try 0, 1,
2 (0 - no variants, 1 - either coding or regulatory variants, 2 - both coding and regulatory
variants) as weights.

**Response 3.3:** Thank you so much for your comments on this. Variants in coding region,
promoter region and UTR region all possess the potentials for affecting phenotypes in
plants. Variants in coding region usually functions by yielding altered protein, while
variants in promoter region and UTR region are more likely to alter gene expression
level.

According to your suggestion, we have altered the weights of sequence variants
identified by RNA-Seq among coding region and UTR regions (0 - no variants, 1 -
either coding or UTR variants, 2 - both coding and UTR variants). Considering larger
260 fold changes did not indicate larger possibility of candidate genes, we circumspectly
selected the weights of DEGs as 0 vs. 1 finally. Indeed, altered weights of sequence
variants elevated the ranks of candidate genes to some degree. We have revised this
manuscript and results have been shown in Figure 3 and Supplementary Figure 2 (in
revised manuscript), respectively.

It was good that the authors were able to obtain EMS mutants or CRISPR/Cas9 lines to
perform functional validation of candidate genes. However, although the validated
genes through mutants or edited-lines did affect TBN, it doesn't necessarily mean they
are the actual functional genes. They could also be genes of pleiotropy. A pleiotropic
gene might be mediated by distinct cis-regulatory variants. Have you investigated other

typical agronomic traits for these genes? Such supporting data would be more
convincing to claim they are actual TBN genes.

**Response 3.4:** Thank you so much for pointing out this. Indeed, many known
functional genes usually showed pleiotropic phenotypes. In our manuscript, genes
affecting TBN in maize were defined as TBN-related positive genes with potential
pleiotropic phenotypes.

**Response Figure 1.** Typical plant architecture traits of CRISPR-edited mutants and wild
type counterparts of *ZmKinesin* and *ZmHD-ZIP120*. (A-B), Five typical plant
architecture traits of *ZmKinesin*. (C-D), Five typical plant architecture traits of *ZmHD-*
*ZIP120*. PH, plant height; EH, ear height; LL, leaf length of the first ear; LW, leaf width
of the first ear; LAE, leaf angle above the first ear. *P* values were determined by
Student's t-tests. **P* < 0.05, ***P* < 0.01, ****P* < 0.001, ns, not significant.

According to your suggestion, we investigated several typical plant architecture traits
of seven cloned genes, including plant height (PH), ear height (EH), leaf length of the
primary ear (LL), leaf width of the primary ear (LW) and leaf angle above the primary
ear (LAE). For *ZmKinesin*, there are no significant differences in PH, LL, LAE and LW
between CRISPR-edited mutants and wild type counterparts except significant increase

of EH in *ZmKinesin*-KO#2 (Response Figure 1). For *ZmHD-ZIP120*, compared with
 wild type counterparts, CRISPR-edited mutants exhibited slightly increased ear height,
 significantly decreased leaf length and leaf width (Response Figure 1).

For six EMS materials, we also investigated five typical plant architecture traits
 referred above (Response Figure 2). For *ZmKinesin*, no significant difference was
 occurred between EMS materials and wild type counterparts. For *ZmOXR*, there are
 significant difference of plant height, ear height and leaf width between EMS materials
 and wild type counterparts. For *NS2*, there are significant difference of plant height, ear
 height and leaf length between EMS materials and wild type counterparts. For *LRS1*,
 slightly decreased plant height was occurred between EMS materials and wild type
 counterparts. For *ZmPRP4K*, there are significant difference of plant height, ear height
 and LAE between EMS materials and wild type counterparts. For *ZmVAMP*, only leaf
 length exhibited significant difference between EMS materials and wild type
 counterparts.

Response Figure 2. Typical plant architecture traits of EMS materials and wild type
 counterparts of six cloned genes. (A) Plant height. (B) Ear height. (C) Leaf length of
 the primary ear. (D) Leaf width of the primary ear. (E) Leaf angle above the primary
 ear. *P* values were determined by Student's *t*-tests. **P* < 0.05, ***P* < 0.01, ****P* < 0.001,
 308 ns, not significant.

The authors used a whole section (Line 281-303) to stress the robustness of QTG-Miner.

However, the evidences are not strong enough to claim the robustness. First, as stated

above, the validated genes may not be the actual TBN genes. Second, except for one
QTL (qTBN3-1) which has three mutation alleles (one EMS mutant and two Cas9-
edited lines), the other six validated genes all have only one mutation allele, which is
generally not enough (at least two mutation alleles or even overexpression alleles
should be used) for functional validation. Third, it is important to evaluate traits of
targeted genes across multiple years and multiple environments. From the TBN
phenotypes of WT and mutants in Fig 3 and SI Fig 1, it looks like at least four QTLs
(qTBN3-1, qTBN4-2, qTBN8-3, and qTBN10-2) have genotype-by-environment
interaction effects. Mutants at qTBN8-3 and qTBN10-2 didn't even show significant
TBN effects in one of the environments, and TBN was only evaluated at one
environment one year for Cas9-edited lines. Therefore, I suggest the authors weaken
the description of this section, especially avoid using percentages as the denominators
are so small (7/12, 5/8, 2/4). Correspondingly, Figure 4 is better as a supplemental
figure rather than a main figure.

**Response 3.5:** Thank you so much for your suggestions on this. According to your
suggestions, we have revised our manuscript as follows:

“Further, we summarized the success rate of fine mapping and cloning of TBN QTLs
with different effects. In field tests from at least two seasons/locations, 7 of the 12 TBN
candidate genes highlighted by QTG-Miner were validated (Supplementary Figure 4A).
To test the performance of QTG-Miner for minor-effect QTLs, we classified the seven
verified TBN QTLs based on the strength of their effects. These seven TBN QTLs have
logarithm of the odd (LOD) values ranging from 3.3 to 8.4 and effects of 4.6–14.4%.
Of the 12 TBN QTLs, 5 QTLs whose LOD values below 5, and 2 QTLs whose LOD
values larger than 5, were successfully verified, respectively (Supplementary Figure
4B). Taken together, these results indicate that QTG-Miner exhibited good performance
for the fine mapping of TBN QTLs with both major and minor effects.” (Lines 258-
268)

Compared to traditional positional cloning, one weakness of the strategy proposed in

this study is that it can't identify the causal genetic polymorphisms for QTLs. Knowing
the causal variants of genes can largely help us understand the evolutionary path for
TBN during maize domestication and improvement. The authors should include this
important aspect in the discussion.

**Response 3.6:** Thank you so much for pointing out this. Here, we indeed did not verify
causal genetic polymorphisms for identified QTGs. Several other methods could help
349 us to assure the causal genetic polymorphisms, for example candidate gene association
analysis, haplotype analysis and other molecular experiments. Following your
suggestions, we discussed this in the Discussion part as follows:

“In addition, QTG-Miner could not identify the causal genetic polymorphisms for
QTGs directly. Several other methods could aid to dissect the causal genetic
polymorphisms. Full-length resequencing of functional gene between two parents,
candidate gene association analysis and haplotype analysis could help to identify the
functional variants, as well as various molecular and genetic experiments.” (Lines 465-
470)

Other points:

Figure 3 - The sequence variant of parents (3D and 3M) is not information for the figure
(the phenotypes are from mutants or gene-edited lines rather than parents), so it is better
to move them to supplemental. I can barely see the difference of TBN in 3E. Do you
have a better tassel picture? The picture of 3H is not representative as compared to the
phenotypic means from 3I. There is about 6 and 8 TBN for WT and Cas9-edited lines
in 3I, while I can only see 4 and 6 in 3H. Please indicate the genotypes in all tassel
pictures and the sample sizes for all statistical tests in Fig 3 and SI Fig 1. The legends
are redundant for the two examples.

**Response 3.7:** Thank you so much for pointing out this. According to your suggestions,
the sequence variants of parents have been moved to Supplementary Figure 3 (in
revised manuscript). In this manuscript, all TBN trait referred includes main axis of
tassel. Following your suggestion, for phenotypic pictures referred above, we have

replaced it with a better and representative tassel picture. Genotypes in all tassel
pictures and the sample sizes for all statistical tests in Figure 3 and Supplementary
Figure 2 have been added. Meanwhile, the redundant legends in Figure 3 have been
removed.

How did the authors define the QTL region for candidate genes? I assume some level
of support interval was used, but it might be not true. I'm sure the homozygous non-
recombinants at target QTL were used for RNA-seq and sequencing. Such information
is unclear until you specify them somewhere in the text.

**Response 3.8:** Thank you so much for your question on this. All QTLs referred here
were identified by our previous study³⁷ (Pan et.al, 2017), and the support intervals were
defined as 1-LOD drop region from the top signal. All RIL materials used in this study
exhibited homozygous genotypes at target QTL. Following your suggestions, the
detailed genotypes of RIL materials across the whole genome were exhibited in
Supplementary Table 2, and graphical presentation of RIL materials were showed in
Supplementary Figure 1. Meanwhile, we have introduced detailed description in
Results as follows:

“The detailed genotype information of 12 pairs of RILs were exhibited and
graphically presented (Supplementary Table 2, Supplementary Figure 1).” (Lines 176-
178)

All of QTL figures shown in the manuscript (Fig 3, SI Fig 1) seem to be schematic
diagram. Real QTL mapping diagram won't have such smooth LOD curve. Please use
the actual LOD curve mapped from the corresponding bi-parental population.

**Response 3.9:** Thank you for your suggestion. We have added the actual LOD curve of
seven QTLs mapped from the corresponding bi-parental population in the revised
manuscript.

Figure 2 - In 2E and 2F, red and blue solid dot indicate positive and negative candidate

genes according to the legends. I'm confused with the meanings of positive and
negative. Why are red and blue alternatively distributed in QTLs?

**Response 3.10:** Thank you so much for pointing this out. In Figure 2E and 2F, all red
and blue solid dots indicated potential candidate genes. To avoid misunderstanding with
positive and negative genes, we have changed the color of solid dot for each QTL.

SI Table 3 - It is surprising that the overlap of DEGs and genes with sequence variants
is not very high. Among the seven validated candidate genes, four genes have sequence
variants, one has DE, one has both, and one has neither. Can you explain this? Are there
differential expression for these genes in WT vs. mutants or CRISPR/Cas9 lines?

**Response 3.11:** Thank you so much for pointing out this. In this manuscript, RNA-Seq
was conducted for identification of DEGs and sequence variants. Only sequence
variants in genic regions (including coding region and UTR regions) could be detected
using RNA-Seq data. For the seven validated candidate genes, the detailed sequence
variants and gene expression were showed in the following two Response Tables, as
well as Supplementary Table 4. As we can see, three genes showed both differential
expression and sequence variants (*Zm00001d042795*, *Zm00001d052598* and
*Zm00001d020804*), and other four genes occur sequence variants only. Besides, a 3Kb
InDel occurred in the promoter region of *Zm00001d052598* among two parents.

Response Table 1. Sequence variants among seven identified genes

QTL	Gene_ID	Variant
qTBN3-1	Zm00001d042795	3_prime_UTR_variant&5_prime_UTR_variant&splice_donor_variant&stop_gained
qTBN4-1	Zm00001d052219	3_prime_UTR_variant&5_prime_UTR_variant&stop_gained
qTBN4-2	Zm00001d052598	conservative_inframe_deletion
qTBN7-1	Zm00001d020804	3'-UTR variants
qTBN8-2	Zm00001d012295	3_prime_UTR_variant&5_prime_UTR_variant&splice_region_variant&stop_gained
qTBN8-3	Zm00001d012452	3_prime_UTR_variant&5_prime_UTR_variant&splice_region_variant&stop_gained
qTBN10-2	Zm00001d025939	3_prime_UTR_variant&5_prime_UTR_variant&splice_region_variant&stop_gained

Response Table 2. Detailed information of DEGs among seven identified genes

QTL	gene_id	FPKM_1	FPKM_2	p_value	q_value	significant
qTBN3-1	Zm00001d042795	3.51954	19.4416	0.00335	0.042682	yes
qTBN4-2	Zm00001d052598	0	2.616	0.0001	0.0098	yes
qTBN7-1	Zm00001d020804	35.79	55.956	0.0035	0.0441	yes

Meanwhile, we also detected the expression levels of seven genes in WT vs. mutants or CRISPR/Cas9 lines. For six genes, there were significant expression differences between WT and EMS-mutagenized mutants (Supplementary Table 9). For CRISPR/Cas9 lines, *Zm00001d042795*, there were no significant expression differences between WT and CRISPR-edited mutants, while the expression of *Zm00001d020804* between WT and CRISPR-edited mutants displayed significant differences (Response Figure 3). It is reasonable that there were no significant expression differences between WT and EMS-mutagenized mutants, because EMS-mutagenized mutants are often inactivated more likely by altering protein structure rather than gene expression.

Response Figure 3. Relative expression of CRISPR-edited mutants and wild type counterparts of *ZmKinesin* and *ZmHD-ZIP120*. *P* values were determined by Student's t-tests. **P* < 0.05, ***P* < 0.01, ****P* < 0.001, ns, not significant.

Figure 1 - What is the scale for the 2-4mm tassel?

**Response 3.12:** Thank you for your question. We have added the scale in Figure 1.

SI Table 2 - The sample name of Zm00001d012295 was incorrect. It should be 2295mu,
2295wt. By the way, what is the genetic background of the mutants? They also have
much higher percentage of mapped reads than RILs.

**Response 3.13:** Thank you so much for pointing this out. We have corrected it. The
background of EMS-mutagenized materials is B73, while these RILs were constructed
using 11 different inbred lines shown in Supplementary Table 1.

SI Table 4 - Why is there a subset of genes not available for machine learning?

**Response 3.14:** Thank you so much for pointing out this. For machine learning used in
this study, genes with missing attributes were not permitted, so a small subset of genes
was excluded. These missing genes are likely pseudo-genes because they were not
expressed in most tissues.

Line 33: The full names for lrs1 should be given here for the first appearance.

**Response 3.15:** Thank you so much for pointing this out. We have added the full name
for its first appearance.

Line 131: To show the genetic background at each TBN QTL, can you provide a
supplemental graph with genotypic information of ten chromosomes for RILs used for
each QTL and indicate the target QTL in the graph?

**Response 3.16:** Thank you so much for pointing out this. Following your suggestion,
we have exhibited the genetic backgrounds of each TBN QTL in Supplementary Figure
1 and Supplementary Table 2.

Line 222 - There are no CRISPR/Cas9 materials in SI Table 6.

**Response 3.17:** Thank you so much for pointing this out. We added the information of
CRISPR/Cas9 materials in Supplementary Table 7.

Line 240 - “resulting in” is a strong word. Change the words as there is no experiment
done in vivo or in vitro to verify that the 8-bp InDel is the actual variant affecting the
transcript level.

**Response 3.18:** Thank you so much for pointing out this. We have rephrased this word.

Line 311 - SI Table 8 only has source and target for network. Where is the information
of these 1,012 DEGs, such as transcript levels in WT and mutants, fold change, q value,
etc.?

**Response 3.19:** Thank you so much for pointing out this. Following your suggestions,
we have revised the table (Supplementary Table 9 in revised manuscript), which
displayed the detailed information of these 1,349 (not 1,012) DEGs identified from six
EMS-mutagenized materials, as well as 957 target genes of ZmHD-ZIP120 using
tsCUT&Tag method.

Line 530 - Which developmental stage was for the 2-4mm immature tassel? Does the
RIL pair for each QTL have difference?

**Response 3.20:** The 2-4mm immature tassels are likely to be in the early stage of
inflorescence development. In this stage, inflorescence meristem (IM) initiates
determinate axillary meristem (AM) and several lateral branch meristems (BMs), which
are eventually responsible for the formation of long branches observed in mature tassels.
Developmental stages of the RIL pair for each QTL have no obvious difference, which
maximized the rigor of the experiment.

Line 538 - Can you provide details regarding CRISPR/Cas9 experiments, such as
gRNA, transformation method, etc.?

**Response 3.21:** Thank you so much for pointing this out. The gRNA information of
*ZmKinesin* and *ZmHD-ZIP120* have been exhibited in Supplementary Table 7. The
detailed information for CRISPR/Cas9 experiments has been introduced in the Methods
as follows:

**“Knockout of *ZmKinesin* and *ZmHD-ZIP120* by CRISPR/cas9 system**

The CRISPR/Cas9 constructs for *ZmKinesin* and *ZmHD-ZIP120* were generated.
The specific guide-RNAs designed for *ZmKinesin* and *ZmHD-ZIP120* were
incorporated into the pCPB-ZmUbi-hspCas9 vector, respectively (Figure 3 and
Supplementary Table 7)⁷¹. All constructs were introduced into the *Agrobacterium* strain
EHA105 and transformed into the immature embryo of the maize inbred line KN5585
through *Agrobacterium*-mediated transformation. CRISPR/Cas9 knockout experiments
of *ZmKinesin* and *ZmHD-ZIP120* were conducted by Wimi Biotechnology Co., Ltd.
(Changzhou, China).

The target regions of *ZmKinesin* and *ZmHD-ZIP120* were amplified from KN5585
and corresponding transgenic lines and sequenced to identify the mutations. For
*ZmKinesin*, we obtained two independent homozygous knockout lines named
*Zmkinesin-KO#1* and *Zmkinesin-KO#2*. For *ZmHD-ZIP120*, we obtained one
independent homozygous knockout line *Zmhd-zip120-KO#1* (Figure 3 and
Supplementary Table 7).” (Lines 647-661)

Line 549 - For RCBD, how many blocks (replicates) and how many treatments
(genotypes) within each block? How did you correct TBN phenotype for the
experimental design?

**Response 3.22:** Thank you so much for pointing out this. Each mutant plot was planted
in replicate with a neighboring wild-type control plot. Two replicates were used for
these phenotyping trials.

Line 556 - I believe you were sequencing mRNA rather than total RNA.

**Response 3.23:** Thank you so much for your reminder on this. We have corrected it.

Line 561 - Did you use default parameters for Hisat2 and Cuffdiff? What is the number
of mismatches and the maximum number of multi-mapped positions allowed? Did you
filter low read count before performing DE test? There are many genes with 0

expression level in one genotype, but you really don't know if those genes are actually
not expressed or just because the sequencing library size doesn't reach the cutoff for
them to be detected.

**Response 3.24:** Thank you so much for pointing out this. The low-quality reads had
been filtered before conducting sequence alignments and DEGs identification. We used
default parameters of Hisat2 and Cuffdiff For RNA-Seq data analysis. The default
maximum number of mismatches allowed is 10, and the maximum number of multi-
mapped positions allowed is 10. For Cuffdiff, the minimum number of alignments in a
locus for DE test is 10 (default). Considering the situation of multi-mapped positions,
we used a parameter (cuffdiff -u/--multi-read-correct) to accurately measure reads
mapping to multiple loci in the genome.

All sequenced samples referred in this study obtained at least 6-Gb clean data, which
is sufficient for gene expression analysis [Conesa, A et al. A survey of best practices for
RNA-seq data analysis. *Genome Biology*, (2016)]. On the one hand, the analysis results
of Cuffdiff showed the test status of input samples for each gene, which included OK
(test successful), NOTEST (not enough alignments for testing), LOWDATA, HIDATA,
or FAIL. Only successfully tested genes (status OK) were conducted expression
analysis. On the other hand, we would further verify the sequence variants among gene
regulatory region if one gene was verified as candidate gene.

Reviewer #4 (Remarks to the Author):

The manuscript has several key topics: 1. The development of QTG-Miner, a
method/algorithm for rapid cloning of candidate genes, 2. The genetics and mechanistic
understanding of tassel branch number (TBN) in maize, 3. the selection history of TBN.

The manuscript presents an impressive data set and relevant analyses to address the
above mentioned topics. It is well written.

The objectives of the manuscript are well presented in the introduction. The main focus
is on the development of QTG Miner. The authors claim that QTG Miner is a novel
method to speed up gene cloning for quantitative traits and outperform conventional
methods with respect to time and costs.

The authors present 12 QTL regions controlling TBN identified in an earlier study. In
seven of them QTG Miner identified causal genes that could be validated in EMS or
CRISPR mutants. In addition they present a molecular network underlying TBN and
analyze the selection pathways of TBN in male and female heterotic maize groups.

**Response:** Thank you very much for your evaluation on our manuscript.

The manuscript would have profited from a more focused presentation of either the
QTG Miner OR the mechanistic insights on TBN. By trying to cover both topics it has
not been possible to cover either of the topics with sufficient depth.

**Response 4.1:** Thank you very much for your comment on this. In this study, we
developed an alternative approach, QTG-Miner, for rapid cloning of QTGs. Meanwhile,
we hoped that it could help us to understand the genetic and molecular basis of
important agricultural traits. Therefore, we constructed an integrated TBN network, and
uncovered the potential relationships among these genes in TBN network, as well as
selection signatures during modern maize breeding. In a word, our research provides
not only an alternative method for QTL rapid cloning, but also a formwork for
systematically dissecting genetic and molecular basis of important agricultural traits at
the omics network perspective. Anyway, thank you again for your constructive
comments.

With respect to QTG Miner quite a few important points were not covered:

The procedure needs a lot of prior information, eg mapped QTL and known genes
regulating the trait. In the last paragraph of the discussion this is briefly mentioned. The
list of positive genes for TBN in maize is long and this is not the case for most

quantitative traits and crops. The efficiency of the method should have been tested with
a smaller number of positive genes to determine a threshold and report on the required
prior information.

**Response 4.2:** Thank you so much for your suggestion on this. QTG-Miner was
developed as an alternative for accelerating the procedure of fine mapping. What QTG-
Miner could do is to prioritize candidate genes driven by AI methods on the multi-omics
network big-data. Similar to fine-mapping, previous biological knowledge about the
genes would be a bonus for candidate gene selection and validation in QTG-Miner.

As we know, more positive genes could largely accelerate novel gene cloning than
limited positive genes. Indeed, for most crops and quantitative traits, only a few
functional genes have been cloned. A series of studies have verified that many causal
genes underlying important agricultural traits, for example plant height, flowering time,
and tassel-related traits, are selection target traits in evolution and domestication and
are somehow conserved across species. Therefore, an alternative strategy for novel gene
cloning is using the known homologous genes cloned in model plants and crops, such
as Arabidopsis, rice and maize. Following your suggestions, we added more discussion
in the Discussion part as follows:

“Meanwhile, a series of studies have verified that many causal genes underlying
important agricultural traits, for example plant height, flowering time, and tassel-related
traits, are selection targets in evolution and domestication, and are somehow conserved
across species^{66, 67, 68.}” (Lines 530-533)

An important parameter to judge the performance of machine learning is AUC value.
Empirically, machine learning algorithms performed well when the AUC values are
larger than 0.8 or 0.85. Following your direction, we also tested the performance of
QTG-Miner with several smaller numbers of positive genes. Five different gradients of
positive gene numbers (10, 15, 20, 25, 32) were set to determine a threshold. For the
former four gradients, positive genes were randomly sampled three times and

conducted data training. Five mean AUC values were showed in Response Figure 4.
Logarithmic function was used in curve fitting. We can see, when the number of positive
genes was 15, the average AUC values could reach up to 0.85, which means that 15
positive genes was sufficient for machine learning.

Response Figure 4. AUC values originated from five gradient positive datasets. Black
solid dot indicated mean AUC values. Curve was fitted using logarithmic function.

One of the cloned genes (TBN7) was not the top hit in the SD and ML analyses (no 3
and 5). The authors did not describe how they converged on the causal gene from the
list the algorithms provided even though others had higher scores.

**Response 4.3:** Thank you so much for pointing this out. Following the suggestion of
reviewer#3 (sequence variants in regulatory region should also be considered and
functioned as weights), we reanalyzed our data and got new rank of causal gene
underlying *qTBN7-1*, and ranked 1st and 3rd for SD and ML, respectively.

Meanwhile, QTG-Miner is an alternative to fine-mapping. Even conventional map-
based cloning can usually narrow-down to several genes, then elect one strong
candidate gene for functional validation based on the prior biological knowledge.
Additionally, the top 1st candidate might be the real functional gene conferring the target
trait, however, it might not have any functional variation in the QTL between the
mapping population.

It was not reported how many genes the 12 QTL regions comprised, how many

candidates per QTL region were selected and how many failures in the validation were
observed (eg how many mutants were analyzed in total to validate the seven genes).

**Response 4.4:** Thank you so much for pointing this out. For seven verified QTLs, the
detailed gene number was showed in Figure 3 and Supplementary Figure 2, for example
totally 53 genes in *qTBN3-1*(Figure 3B). For each QTL, we only nominated one gene
as candidate gene. Among 12 TBN QTLs, seven could be verified by EMS materials or
CRISPR/Cas9 lines.

What happened in the 5 QTL regions where no causal gene could be identified?

**Response 4.5:** Thank you for your question. QTG-Miner can prioritize candidate genes
for each QTL like the procedure of fine-mapping. Researchers could elect the strongest
candidate for functional validation by integrating the prior background knowledge.
However, this procedure is complex, similar to conventional fine-mapping, which
narrows down the QTL to a ten-of-kb region with about 10 genes. In our study, we
nominated only one gene as candidate for each QTL given the huge workload. Five
candidate genes from corresponding QTLs could not be validated using EMS materials,
which might be caused by functional redundancy of homologous genes, wrong
nominations of candidate genes, or other factors.

It was not described how the results from the SD and ML analyses differed with respect
to the success criterion. The introduction of a new method requires a more detailed
coverage with respect to parameter settings, specificity and sensitivity of the method.

**Response 4.6:** Thank you very much for your comments. It is hard to judge the
performance between the SD and ML methods, because of limited verified QTLs. So,
QTG-Miner is a new strategy like fine-mapping to provide two alternative methods for
accelerating functional gene cloning. Users could take both results from SD and ML
into consideration.

It would have been interesting to see if in some QTL regions the algorithm was more
successful than in others.

**Response 4.7:** Thank you so much for pointing out this. QTG-Miner exhibited equally
good performances for both QTLs with major effect and minor effect, even though only
limited QTLs preliminarily support this conclusion. From a perspective of gene
function, QTG-Miner was more likely to identify genes belonging to several known
pathways with many known positive genes, for example meristem maintenance and
determinacy, because of the tight connections between candidate genes and positive
genes.

How was the list of negative genes chosen for the ML algorithm, why different numbers
of positive genes in SD and ML?

**Response 4.8:** Thank you for your question on this. We should have clarified it clearly
previously. Negative genes referred here had never been reported affecting tassel-
related traits, and were screened according to functional annotations, expression
profiles, including some classical maize genes
(https://genomevolution.org/wiki/index.php/Classical_Maize_Genes), house-keeping
genes (for example actin-related genes) and genes of tissue-specific expression. It worth
notice that there is still the possibility of false-negative genes because of limited
understanding of gene functions. Accordingly, we have added the functional
annotations of all these positive and negative genes in Supplementary Table 5.

For machine learning used in this study, genes with missing attributes (without
detectable expression-level) were removed, so a small subset of genes was excluded.

How was the integrated network map built? The methods section should be more
detailed on the chosen parameters.

**Response 4.9:** Thank you so much for your question on this. The integrated multi-
omics network map was constructed in our precious study by Han et al.³⁷, and the
detailed information of map construction was described in this study [Han L, *et al.* A
multi-omics integrative network map of maize. *Nature Genetics*, (2023)].

A more detailed presentation of the results of the DEG analysis would have been useful.
**Response 4.10:** Thank you for your suggestion. The detailed information of DEGs
identified from EMS materials have been showed in Supplementary Table 9.

In the discussion the authors claim that cost and time can be saved with QTG Miner.
That is only the case if a lot of preliminary work has been done on the trait as is the
case for TBN. A more critical discussion on what it takes and how QTG Miner can be
combined with classical methods generating recombinants would have been useful. The
workflow in Figure 1 is not really that novel as it is quite similar to what is done in
todays mapping projects without prior information on candidates.

**Response 4.11:** Thank you for your comments on this. We believe that all the plants are
research models with tons of data since we are in the era of biological big-data
nowadays. So, our QTG-Miner will have a wide potential usage in plants. Following
your suggestions, we twisted QTG-Miner as an alternative method for QTL mapping
and cloning in the discussion part.

“Most agronomic traits such as TBN are quantitative and controlled by many QTLs
with major or minor effects. Conventional methods employ time-consuming and labor-
intensive procedures, and only relying on conventional methods for QTGs cloning is
inefficient and challenging, especially when dissecting such QTLs with minor effects
or represented by rare alleles^{1, 8}. Combing QTG-Miner with conventional methods for
QTGs cloning displayed obvious advantages for QTGs cloning. First, by integrating
QTG-Miner with conventional methods, both major- and minor-effect QTLs could be
rapidly cloned. A large number of recombinants enabled it possible that the selection of
materials segregating for a single QTL alleviated potential interference of unlinked and
background QTLs to a great extent, which helped ensure the accuracy of the identified
DEGs and variants. Second, with the help of various populations constructed using
conventional methods, QTG-Miner can achieve functional gene cloning rapidly and
efficiently when the genetic background of the population is properly controlled.
Population types can be biparental or multiparental, including selfing and backcross

populations, derived RILs, backcross inbred lines, residual heterozygous lines, nested
association mapping populations, or multiparent advanced generation intercross
populations. Third, integrating QTG-Miner with conventional methods could achieve
large-scale gene cloning in a rapid and efficient manner. With the help of QTG-Miner,
iterative and tedious fine mapping could be alleviated, and batch cloning the functional
genes could be achieved at a time. Therefore, integrating QTG-Miner with conventional
methods is a powerful strategy for accelerating QTGs cloning.” (Lines 439-460)

It would have been interesting to learn if prior information from related species would
have been useful (eg include them in the list of positive genes)

**Response 4.12:** Thank you for your suggestion on this. In this manuscript, there are
many positive (TBN-related) genes having been cloned in maize, so we did not take
homologous genes into consideration. For other species and agricultural traits without
enough positive genes, prior information from related species could be taken into
consideration. Following your suggestions, we discussed this in the last paragraph of
Discussion part.

“Meanwhile, a series of studies have verified that many causal genes underlying
important agricultural traits, for example plant height, flowering time, and tassel-related
traits, are selection targets in evolution and domestication, and are somehow conserved
across species^{66, 67, 68}. Therefore, positive gene datasets can be obtained by identifying
putative orthologs to known causal genes in model plant species. Thus, QTG-Miner has
the potential to be widely applied to various species and traits.” (Lines 530-536)

With respect to selection analyses and molecular networks:

Are these sections essential to introduce QTG Miner?

**Response 4.13:** Thank you so much for pointing out this. We hoped that our research
would provide not only an alternative method for QTL rapid cloning, but also a
formwork for systematically dissecting genetic and molecular basis of important

agricultural traits from omics network aspect. Anyway, thank you again for your
constructive comments.

Isn't it counter-intuitive that most of the TBN genes are co-directional in the male and
female heterotic pools?

**Response 4.14:** Thank you so much for pointing out this. Previously, Li et al identified
589 co-directional selected genes among modern hybrid maize breeding⁴⁵ (Li C, et al.,
Nature Genetics, 2022). In our study, 88 genes in TBN network were overlapped with
co-directional selected genes, the proportion of which in total 4278 TBN network genes
is only about 2%, which indicated only a small proportion of TBN genes functioned in
co-directional selection of TBN trait during modern hybrid maize breeding (Figure 6A
and 6B in the revised manuscript). To our knowledge, the observation that some TBN
genes are co-directional in the male and female heterotic pools might be due to the
background suppression of the other TBN functional genes.

Reviewers' Comments:

Reviewer #1:

Remarks to the Author:

Nice work addressing my comments!

Reviewer #2:

Remarks to the Author:

The authors have done their utmost best to address all issues, concerns and questions. To my opinion this manuscript can be accepted for publication.

Reviewer #3:

Remarks to the Author:

The revised version of the manuscript by Wang et al. has much improved. I'm glad to see that altering the weights of sequence variants could help improve the ranks for all candidate genes. In the response, the authors described using 0, 1, 2 as weights but 0, 0.5, 1 in the main text. Please be sure which one is correct.

The last section of Results is confusing. What is the definition of convergent selection and why is there only convergent selected genes only in MHGs but not in FHGs? Figure 6A-C show that convergent selected genes take a large proportion similar to co-directional selection genes, as well as similar enrichments compared to background and similar pathway enrichments. Therefore, I don't understand why the authors only picked up co-directional selection genes as a major conclusion and used it as the second part of the title for the manuscript. I feel like the focus of this manuscript should be QTG-Miner, a title such as "QTG-Miner aids rapid cloning of quantitative trait loci for tassel branch number in maize" seems simpler and more direct to me.

Others:

Figure 2F: Define the gray dotted line. How is the cutoff calculated? Why is Figure 2E without a unique cutoff like 2F? In 2E, each QTL seems to exclude genes by different criterion.

Figure 3K: The title of y-axis is missing.

Fine mapping is generally involved in identification of causal variants. To avoid confusion, delete "fine mapping and" in line 258 and change "fine mapping" in line 267 to "cloning".

Line 318: "infer" rather than "inter".

Line 443: "Combining" rather than "Combing".

Reviewer #4:

Remarks to the Author:

Thank you for submitting a thoroughly revised manuscript.

The authors have addressed the reviewers' comments in a detailed and clear way. They added the information requested by reviewers. The interpretation of the data has been expanded and improved. The relevance of QTG-Miner with respect to other gene cloning methods has been discussed and clarified.

**Point-by-point response to the reviewers' comments**

We sincerely thank all reviewers for their constructive comments and suggestions.
These comments greatly help us to improve the manuscript. We have tried our best to
conduct all the experiments as suggested, addressed the concerns from reviewer 3, and
revised the manuscript accordingly. Our responses to reviewers' comments are in blue
below, as well as in the revised manuscript.

**REVIEWERS' COMMENTS**

Reviewer #1 (Remarks to the Author):

Nice work addressing my comments!

**Response:** Thank you so much for your constructive comments and suggestions, as
well as great efforts paid in reviewing our manuscripts.

Reviewer #2 (Remarks to the Author):

The authors have done their utmost best to address all issues, concerns and questions.
To my opinion this manuscript can be accepted for publication.

**Response:** Thank you so much for your critical comments and final approval, as well
as great efforts paid in reviewing our manuscripts.

Reviewer #3 (Remarks to the Author):

The revised version of the manuscript by Wang et al. has much improved. I'm glad to
see that altering the weights of sequence variants could help improve the ranks for all
candidate genes. In the response, the authors described using 0, 1, 2 as weights but 0,
0.5, 1 in the main text. Please be sure which one is correct.

**Response 3.1:** Thank you very much for pointing this out. We sincerely apologize for
this typo in the Response file. The correct weights of sequence variants are 0, 0.5, 1 in
the main text.

The last section of Results is confusing. What is the definition of convergent selection
and why is there only convergent selected genes only in MHGs but not in FHGs? Figure
6A-C show that convergent selected genes take a large proportion similar to co-
directional selection genes, as well as similar enrichments compared to background and
similar pathway enrichments. Therefore, I don't understand why the authors only
picked up co-directional selection genes as a major conclusion and used it as the second
part of the title for the manuscript. I feel like the focus of this manuscript should be
QTG-Miner, a title such as "QTG-Miner aids rapid cloning of quantitative trait loci for
tassel branch number in maize" seems simpler and more direct to me.

**Response 3.2:** Thank you very much for your question on this. Convergent selection
indicates convergent allele frequency changes in the FHGs or MHGs by **Li et al.** In
their study, they identified convergent selected genes only in MHGs but not in FHGs.
The possible reason they provided might be that MHGs seem to have experienced more
intensive artificial selection than the FHGs during modern hybrid maize breeding.

Indeed, convergent selected genes take a large proportion similar to co-directional
selection genes (Fig. 6a-c), but only two significant enrichment terms were identified
(reciprocal DNA recombination, reciprocal meiotic recombination) (Supplementary
Data 22 in revised version). Therefore, we picked up co-directional selection genes as
a major conclusion.

We do agree with your opinion on this. We twisted the title of our manuscript as
"*QTG-Miner aids rapid dissection of the genetic base of tassel branch number in*
*maize*"

**Reference:** Li C, et al. Genomic insights into historical improvement of heterotic
groups during modern hybrid maize breeding. *Nature Plants* 8, 750-763 (2022).

Others:

Figure 2F: Define the gray dotted line. How is the cutoff calculated? Why is Figure 2E
without a unique cutoff like 2F? In 2E, each QTL seems to exclude genes by different
criterion.

**Response 3.3:** Thank you very much for pointing this out. According to your suggestion,
we have defined the gray dotted line in the revised manuscript. In Fig. 2f, the cutoff is
1/3, which is defined according the formula below:

$$P_w = \frac{(P_n + P_d + P_v)}{3}$$

P_n is the probability obtained from the NeuralNet algorithm, ranging from 0 to 1. P_d
and P_v are the weights of the DEG and variants, respectively. If a gene was identified
as a DEG, its P_d value equals 1, and otherwise is equal to 0. For any excluded genes, its
P_d and P_v values are equal to 0, even though its P_n value might be as large as 1. Therefore,
the cutoff is 1/3, which can discriminate positive genes from negative ones.

In Fig. 2e, we could not provide a unique cutoff for each QTL region, because of the
uncertainty of the largest SD_g value across different QTL regions. Therefore, we
selected a compromise approach for the cutoff.

Figure 3K: The title of y-axis is missing.

**Response 3.4:** Thank you for your reminder on this. We have added the title of y-axis
in Fig. 3k.

Fine mapping is generally involved in identification of causal variants. To avoid
confusion, delete “fine mapping and” in line 258 and change “fine mapping” in line 267
to “cloning”.

**Response 3.5:** Thank you for your suggestion on this. We revised it as you suggested.

Line 318: “infer” rather than “inter”.

**Response 3.6:** We sincerely apologize for this typo. We corrected it in the revised
manuscript. Thank you so much.

Line 443: “Combining” rather than “Combing”.

**Response 3.7:** We sincerely apologize for this typo. We corrected it in the revised
manuscript. Thank you very much.

Reviewer #4 (Remarks to the Author):

Thank you for submitting a thoroughly revised manuscript.

The authors have addressed the reviewers' comments in a detailed and clear way. They
added the information requested by reviewers. The interpretation of the data has been
expanded and improved. The relevance of QTG-Miner with respect to other gene
cloning methods has been discussed and clarified.

**Response:** Thank you so much for your constructive comments and valuable
suggestions, as well as great efforts paid in reviewing our manuscripts.
